# DECAF: Generating Fair Synthetic Data Using Causally-Aware Generative Networks

**Boris van Breugel**[*]
University of Cambridge
bv292@cam.ac.uk

**Trent Kyono**[*]
University of California, Los Angeles
tmkyono@ucla.edu

**Jeroen Berrevoets**
University of Cambridge
jb2384@cam.ac.uk

**Mihaela van der Schaar**
University of Cambridge
University of California, Los Angeles
The Alan Turing Institute
mv472@cam.ac.uk

## Abstract

Machine learning models have been criticized for reflecting unfair biases in the training data. Instead of solving for this by introducing fair learning algorithms directly, we focus on generating fair synthetic data, such that any downstream learner is fair. Generating fair synthetic data from unfair data— *while remaining truthful to the underlying data-generating process (DGP)* —is non-trivial. In this paper, we introduce DECAF: a GAN-based *fair* synthetic data generator for tabular data. With DECAF we embed the DGP explicitly as a structural causal model in the input layers of the generator, allowing each variable to be reconstructed conditioned on its causal parents. This procedure enables *inference-time* debiasing, where biased edges can be strategically removed for satisfying user-defined fairness requirements. The DECAF framework is versatile and compatible with several popular definitions of fairness. In our experiments, we show that DECAF successfully removes undesired bias and— in contrast to existing methods —is capable of generating high-quality synthetic data. Furthermore, we provide theoretical guarantees on the generator's convergence and the fairness of downstream models.

## 1 Introduction

Generative models are optimized to approximate the original data distribution as closely as possible. Most research focuses on three objectives [1]: fidelity, diversity, and privacy. The first and second are concerned with how closely synthetic samples resemble real data and how much of the real data's distribution is covered by the new distribution, respectively. The third objective aims to avoid simply reproducing samples from the original data, which is important if the data contains privacy-sensitive information [2, 3]. We explore a much-less studied concept: synthetic data fairness.

**Motivation.** Deployed machine learning models have been shown to reflect the bias of the data on which they are trained [4, 5, 6, 7, 8]. This has not only unfairly damaged the discriminated individuals but also society's trust in machine learning as a whole. A large body of work has explored ways of detecting bias and creating fair predictors [9, 10, 11, 12, 13, 14, 15], while other authors propose debiasing the data itself [9, 10, 11, 16]. This work's aim is related to the work of [17]: to generate fair synthetic data based on unfair data. Being able to generate fair data is important because end-users creating models based on publicly available data might be unaware they are inadvertently including

---

[*]Equal contribution

35th Conference on Neural Information Processing Systems (NeurIPS 2021).

Table 1: **Overview of related work for synthetic data.** We organize related work according to our key areas of interest: **(1)** Allows post-hoc distribution changes, **(2)** provides fairness, **(3)** supports causal notion of fairness, **(4)** allows inference-time fairness, **(5)** requires minimal assumptions. We highlight the key contribution, and identify non-neural approaches with "†".

| Model | Reference | (1) | (2) | (3) | (4) | (5) | Goal |
|---|---|---|---|---|---|---|---|
| | | Standard *synthetic data generation* | | | | | |
| VAE | [19] | ✗ | ✗ | ✗ | ✗ | ✓ | Realistic synth. data. |
| GANs | [2, 3, 20, 21] | ✗ | ✗ | ✗ | ✗ | ✓ | Realistic synth. data. |
| | | Methods that *detect and/or remove bias* | | | | | |
| PSE-DD/DR† | [11] | ✓ | ✓ | ✓ | ✗ | ✗ | Discover/Remove bias. |
| OPPDP† | [16] | ✗ | ✓ | ✗ | ✗ | ✗ | Remove bias. |
| DI† | [10] | ✗ | ✓ | ✗ | ✗ | ✗ | Discover/Remove bias. |
| LFR | [22] | ✗ | ✓ | ✗ | ✗ | ✓ | Learn fair representation. |
| FairGAN | [17] | ✗ | ✓ | ✗ | ✗ | ✓ | Realistic and fair synth. data. |
| CFGAN | [23] | ✗ | ✓ | ✓ | ✗ | ✓ | Realistic and fair synth. data. |
| DECAF | (ours) | ✓ | ✓ | ✓ | ✓ | ✓ | Realistic and fair synth. data. |

bias or insufficiently knowledgeable to remove it from their model. Furthermore, by debiasing the data prior to public release, one can guarantee *any* downstream model satisfies desired fairness requirements by assigning the responsibility of debiasing to the data generating entities.

**Goal.** From a biased dataset $\mathcal{X}$, we are interested in learning a model $G$, that is able to generate an equivalent *synthetic* unbiased dataset $\mathcal{X}'$ with minimal loss of data utility. Furthermore, a downstream model trained on the synthetic data needs to make not only unbiased predictions on the synthetic data, but also on real-life datasets (as formalized in Section 4.2).

**Solution.** We approach fairness from a causal standpoint because it provides an intuitive perspective on different definitions of fairness and discrimination [11, 13, 14, 15, 18]. We introduce DEbiasing CAusal Fairness (DECAF), a generative adversarial network (GAN) that leverages causal structure for synthesizing data. Specifically, DECAF is comprised of $d$ generators (one for each variable) that learn the causal conditionals observed in the data. At inference-time, variables are synthesized topologically starting from the root nodes in the causal graph then synthesized sequentially, terminating at the leave nodes. Because of this, DECAF can remove bias at inference-time through targeted (biased) edge removal. As a result, various datasets can be created for desired (or evolving) definitions of fairness.

**Contributions.** We propose a framework of using causal knowledge for fair synthetic data generation. We make three main contributions: i) DECAF, a causal GAN-based model for generating synthetic data, ii) a flexible causal approach for modifying this model such that it can generate fair data, and iii) guarantees that downstream models trained on the synthetic data will also give fair predictions in other settings. Experimentally, we show how DECAF is compatible with several fairness/discrimination definitions used in literature while still maintaining high downstream utility of generated data.

## 2 Related Works

Here we focus on the related work concerned with data generation, in contrast to fairness definitions for which we provide a detailed overview in Section 4 and Appendix C. As an overview of how data generation methods relate to one another, we refer to Table 1 which presents all relevant related methods.

**Non-parametric generative modeling.** The standard models for synthetic data generation are either based on VAEs [19] or GANs [2, 3, 20, 21]. While these models are well known for their highly realistic synthetic data, they are unable to alter the synthetic data distribution to encourage fairness (except for [17, 23], discussed below). Furthermore, these methods have no causal notion, which prohibits targeted interventions for synthesizing fair data (Section 4). We explicitly leave out CausalGAN [24] and CausalVAE [25], which appear similar by incorporating causality-derived ideas but are different in both method and aim (i.e., image generation).

**Fair data generation.** In the bottom section of Table 1, we present methods that, in some way, alter the training data of classifiers to adhere to a notion of fairness [10, 11, 16, 17, 22, 23]. While

these methods have proven successful, they lack some important features. For example, none of the related methods allow for post-hoc changes of the synthetic data distribution. This is an important feature, as each situation requires a different perspective on fairness and thus requires a flexible framework for selecting protected variables. Additionally, only [11, 23] allow a causal perspective on fairness, despite causal notions underlying multiple interpretations of what should be considered fair [13]. Furthermore, only [17, 22, 23] offer a flexible framework, while the others are limited to binary [10, 11] or discrete [16] settings. Xu et al. [23] also use a causal architecture for the generator, however their method is not as flexible—e.g. it does not easily extend to multiple protected attributes. Finally, in contrast to other methods DECAF is directly concerned with fairness of the downstream model—which is dependent on the setting in which the downstream model is employed (Section 4.2). In essence, from Table 1 we learn that DECAF is the only method that combines all key areas of interest. At last, we would like to mention [26], who aim to generate data that resembles a small unbiased reference dataset, by leveraging a large but biased dataset. This is very different to our aim, as we are interested in the downstream model's fairness and explicit notions of fairness.

## 3 Preliminaries

Let $X \in \mathcal{X} \subseteq \mathbb{R}^d$ denote a random variable with distribution $P_X(X)$, with protected attributes $A \in \mathcal{A} \subset \mathcal{X}$ and target variable $Y \in \mathcal{Y} \subset \mathcal{X}$, let $\hat{Y}$ denote a prediction of $Y$. Let the data be given by $\mathcal{D} = \{\mathbf{x}^{(k)}\}_{k=1}^N$, where each $\mathbf{x}^{(k)} \in \mathcal{D}$ is a realization of $X$. We assume the data generating process can be represented by a directed acyclic graph (DAG)—such that the generation of features can be written as a structural equation model (SEM) [27]—and that this DAG is causally sufficient. Let $X_i$ denote the $i^{\text{th}}$ feature in $X$ with causal parents $\text{Pa}(X_i) \subset \{X_j : j \neq i\}$, the SEM is given by:

$$X_i = f_i(\text{Pa}(X_i), Z_i), \forall i \tag{1}$$

where $\{Z_i\}_{i=1}^d$ are independent random noise variables, that is $\text{Pa}(Z_i) = \emptyset$, $\forall i$. Note that each $f_i$ is a deterministic function that places all randomness of the conditional $P(X_i | \text{Pa}(X_i))$ in the respective noise variable, $Z_i$.

## 4 Fairness of Synthetic Data

Algorithmic fairness is a popular topic (e.g., see [13, 28]), but *fair synthetic data* has been much less explored. This section highlights how the underlying graphs of the synthetic and downstream data determine whether a model trained on the synthetic data will be fair in practice. We start with the two most popular definitions of fairness, relating to the legal concepts of *direct* and *indirect* discrimination. We also explore *conditional fairness* [29], which is a generalization of the two. In Appendix C we discuss how the ideas in this section transfer to other independence-based definitions [30]. Throughout this section, we separate $Y$ from $X$ by defining $\bar{X} = X \backslash Y$, and we will write $X \leftarrow \bar{X}$ for ease of notation.

### 4.1 Algorithmic fairness

The first definition is called *Fairness Through Unawareness* (e.g. [31]).

**Definition 1.** *(Fairness Through Unawareness (FTU): algorithm). A predictor $f : X \mapsto \hat{Y}$ is fair iff protected attributes $A$ are not explicitly used by $f$ to predict $\hat{Y}$.*

This definition prohibits *disparate treatment* [28, 32], and is related to the legal concept of *direct discrimination*, i.e., two equally qualified people deserve the same job opportunity independent of their race, gender, beliefs, among others.

Though FTU fairness is commonly used, it might result in *indirect discrimination*: covariates that influence the prediction $\hat{Y}$ might not be identically distributed across different groups $a, a'$, which means an algorithm might have *disparate impact* on a protected group [10]. The second definition of fairness, *demographic parity* [32], does not allow this:

**Definition 2.** *(Demographic Parity (DP): algorithm) A predictor $\hat{Y}$ is fair iff $A \perp\!\!\!\perp \hat{Y}$, i.e. $\forall a, a'$ : $P(\hat{Y} | A = a) = P(\hat{Y} | A = a')$.*

Evidently, DP puts stringent constraints on the algorithm, whereas FTU might be too lenient. The third definition we include is based on the work of [29], related to *unresolved discrimination* [14]. The idea is that we do not allow indirect discrimination unless it runs through *explanatory factors* $R \subset X$. For example, in Simpson's paradox [33] there seems to be a bias between gender and college admissions, but this is only due to women applying to more competitive courses. In this case, one would want to regard fairness conditioned on the choice of study [14]. Let us define this as *conditional fairness*:

**Definition 3.** *(Conditional Fairness (CF): algorithm) A predictor $\hat{Y}$ is fair iff $A \perp\!\!\!\perp \hat{Y}|R$, i.e. $\forall r, a, a' : P(\hat{Y}|R = r, A = a) = P(\hat{Y}|R = r, A = a')$.*

**CF generalizes FTU and DP** Note that conditional fairness is a generalization of FTU and DP, by setting $R = X \backslash A$ and $R = \emptyset$, respectively. In Appendix C we elaborate on the connection between these, and more, definitions.

## 4.2 Synthetic data fairness

Algorithmic definitions can be extended to distributional fairness for synthetic data. Let $P(X), P'(X)$ be probability distributions with protected attributes $A \subset X$ and labels $Y \subset X$. Let $\mathcal{I}(A, Y)$ be a definition of algorithmic fairness (e.g., FTU). Note, that under CF, $\mathcal{I}(A, Y)$ is a function of $R$ as well. We propose $(\mathcal{I}(A, Y), P)$-fairness of distribution $P'(X)$:

**Definition 4.** *(Distributional fairness) A probability distribution $P'(X)$ is $(\mathcal{I}(A, Y), P)$-fair, iff the optimal predictor $\hat{Y} = f^*(X)$ of $Y$ trained on $P'(X)$ satisfies $\mathcal{I}(A, Y)$ when evaluated on $P(X)$.*

In other words, when we train a predictor on $(\mathcal{I}(A, Y), P)$-fair distribution $P'(X)$, we can only reach maximum performance if our model is fair. Note the explicit reference to $P(X)$, the distribution on which fairness is evaluated, which does not need to coincide with $P'(X)$. This is a small but relevant detail. For example, when training a model on data $\mathcal{D}' \sim P'(X)$ it could seem like the model is fair when we evaluate it on a hold-out set of the data (e.g., if we simply remove the protected attribute from the data). However, when we use the model for real-world predictions of data $\mathcal{D} \sim P(X)$, disparate impact is possibly observed due to a distributional shift.

By extension, we define synthetic data as $(\mathcal{I}(A, Y), P)$-fair, iff it is sampled from an $(\mathcal{I}(A, Y), P)$-fair distribution. Defining synthetic data as fair w.r.t. an optimal predictor is especially useful when we want to publish a dataset and do not trust end-users to consider anything but performance.[2]

**Choosing $\mathbf{P(X)}$.** The setting $P(X) = P'(X)$ corresponds to data being fair with respect to itself. For synthetic data generation, this setting is uninteresting as any dataset can be made fair by randomly sampling or removing $A$; if $A$ is random, the prediction should not directly or indirectly depend on it. This ignores, however, that a downstream user might use the trained model on a real-world dataset in which other variables $B$ are correlated with $A$, and thus their model (which is trained to use $B$ for predicting $Y$) will be biased. Of specific interest is the setting where $P(X)$ corresponds to the original data distribution $P_X(X)$ that contains unfairness. In this scenario, we construct $P'(X)$ by learning $P_X(X)$ and removing the unfair characteristics. The data from $P'(X)$ can be published online, and models trained on this data can be deployed fairly in real-life scenarios where data follows $P_X(X)$. Unless otherwise stated, henceforth, we assume $P(X) = P_X(X)$.

## 4.3 Graphical perspective

As reflected in the widely accepted terms direct versus indirect discrimination, it is natural to define distributional fairness from a causal standpoint. Let $\mathcal{G}'$ and $\mathcal{G}$ respectively denote the graphs underlying $P'(X)$ (the synthetic data distribution which we can control) and $P(X)$ (the evaluation distribution that we cannot control). Let $\partial_{\mathcal{G}} Y$ denote the Markov boundary of $Y$ in graph $\mathcal{G}$. We focus on the conditional fairness definition because it subsumes the definition of DP and FTU (Section 4.1). Let $R \subset X$ be the set of explanatory features.

**Proposition 1.** *(CF: graphical condition) If for all $B \in \partial_{\mathcal{G}'} Y$, $A \perp\!\!\!\perp_{\mathcal{G}} B|R$,[3] then distribution $P'(X)$ is CF fair w.r.t $P(X)$ given explanatory factors $R$.*

---

[2]Finding the optimal predictor is possible if we assume the downstream user employs any universal function approximator (e.g., MLP) and the amount of synthetic data is sufficiently large.

[3]Where $\perp\!\!\!\perp_{\mathcal{G}}$ denotes d-separation in $\mathcal{G}$. Here we define $A \perp\!\!\!\perp_{\mathcal{G}} B|R$ to be true for all $B \in R$.

*Proof.* Without loss of generality, let us assume the label is binary.[4] The optimal predictor $f^*(X) = P(Y|X) = P(Y|\partial_{\mathcal{G}'}Y)$. Thus, if $\partial_{\mathcal{G}'}Y$ is d-separated from $A$ in $\mathcal{G}$ given $R$, prediction $\hat{Y} = f^*(X)$ is independent of $A$ given $R$ and CF holds. □

**Corollary 1.** *(CF debiasing) Any distribution $P'(X)$ with graph $\mathcal{G}'$ can be made CF fair w.r.t. $P(X)$ and explanatory features $R$ by removing from $\mathcal{G}'$ edges $\tilde{E} = \{(B \to Y) \text{ and } (Y \to B) : \forall B \in \partial_{\mathcal{G}'}Y \text{ with } B \not\perp\!\!\!\perp_{\mathcal{G}} A|R\}$.*

*Proof.* First note $\tilde{E}$ is the necessary and sufficient set of edges to remove for $(\forall B \in \partial_{\mathcal{G}'}Y, A \perp\!\!\!\perp_{\mathcal{G}} B|R)$ to be true, subsequently the result follows from Proposition 1. □

For FTU (i.e. $R = X \backslash A$) and DP (i.e. $R = \emptyset$), this corollary simplifies to:

**Corollary 2.** *(FTU debiasing) Any distribution $P'(X)$ with graph $\mathcal{G}'$ can be made FTU fair w.r.t. any distribution $P(X)$ by removing, if present, i) the edge between $A$ and $Y$ and ii) the edge $A \to C$ or $Y \to C$ for all shared children $C$.*

**Corollary 3.** *(DP debiasing) Any distribution $P'(X)$ with graph $\mathcal{G}'$ can be made DP fair w.r.t. $P(X)$ by removing, if present, the edge between $B$ and $Y$ for any $B \in \partial_{\mathcal{G}'}Y$ with $B \not\perp\!\!\!\perp_{\mathcal{G}} A$.*

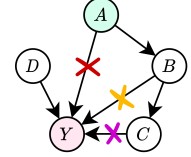

Figure 1: **Edge removal for fairness.** FTU: ✗; DP: ✗✗✗; CF when $R = C$: ✗✗; CF when $B \in R$: ✗.

Figure 1 shows how the different fairness definitions lead to different sets of edges to be removed.

**Faithfulness.** Usually one assumes distributions are faithful w.r.t. their respective graphs, in which case the if-statement in Proposition 1 become equivalence statements: fairness is *only* possible when the graphical conditions hold.

**Theorem 1.** *If $P(X)$ and $P'(X)$ are faithful with respect to their respective graphs $\mathcal{G}$ and $\mathcal{G}'$, then Proposition 1 becomes an equivalence statement and Corollaries 1, 2 and 3 describe the necessary and sufficient sets of edges to remove for achieving CF, FTU and DP fairness, respectively.*

*Proof.* Faithfulness implies $A \perp\!\!\!\perp_{P(X)} B|R \implies A \perp\!\!\!\perp_{\mathcal{G}} B|R$, e.g. [34]. Thus, if $\exists B \in \partial_{\mathcal{G}'}Y$ for which $A \not\perp\!\!\!\perp_{\mathcal{G}} B|R$, then $A \not\perp\!\!\!\perp B|R$. Because $B \in \partial_{\mathcal{G}'}Y$ and $P'(X)$ is faithful to $\mathcal{G}'$, $\hat{Y} = f^*(X)$ depends on $B$, and thus $\hat{Y} \not\perp\!\!\!\perp A|R$: CF does not hold. □

**Other definitions.** Some authors define similar fairness measures in terms of directed paths (cf. d-separation) [11, 14, 18], which is a milder requirement as it allows correlation via non-causal paths. In Appendix C we highlight the graphical conditions for these definitions.

# 5 Method: DECAF

The primary design goal of DECAF is to generate fair synthetic data from unfair data. We separate DECAF into two stages. The training stage learns the causal conditionals that are observed in the data through a causally-informed GAN. At the generation (inference) stage, we intervene on the learned conditionals via Corollaries 1-3, in such a way that the generator creates fair data. We assume the underlying DGP's graph $\mathcal{G}$ is known; otherwise, $\mathcal{G}$ needs to be approximated first using any causal discovery method, see Section 6.

## 5.1 Training

**Overview.** This stage strives to learn the causal mechanisms $\{f_i(\text{Pa}(X_i), Z_i)\}$. Each structural equation $f_i$ (Eq. 1) is modelled by a separate generator $G_i : \mathbb{R}^{|Pa(X_i)|+1} \to \mathbb{R}$. We achieve this by employing a conditional GAN framework with a causal generator. This process is illustrated in Figure 2 and detailed below.

---

[4]If $Y$ is continuous the same result holds, though the "optimal" predictor will depend on the statistic of interest, e.g. mode, mean, median or the entire distribution $f(X, Y) \approx P(Y|X)$.

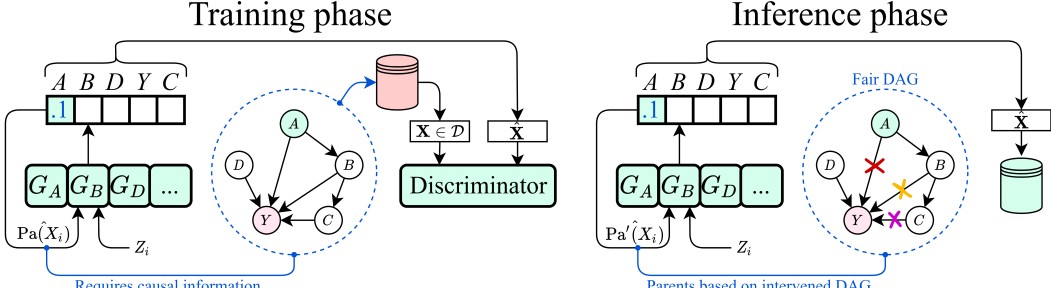

Figure 2: **Architecture.** *Training phase—* Each component in $\hat{\mathbf{X}}$ is generated sequentially as a function (where the function is that component's generator $G_i$) of the component's parents. Parental knowledge is provided by the DAG governing the data. *Inference phase—* As the component-wise generation of the generator network is independent of the DAG governing the data, we can easily replace (or intervene on) the DAG governing parental information. The resulting synthetic data (right) will be governed by the intervened DAG. *FTU is achieved by removing: ✗; DP: ✗✗✗; e.g. CF when $R = C$: ✗✗.*

Features are generated sequentially following the topological ordering of the underlying causal DAG: first root nodes are generated, then their children (from generated causal parents), etc. Variable $\hat{X}_i$ is modelled by the associated generator $G_i$:

$$\hat{X}_i = G_i(\hat{\mathrm{Pa}}(X_i), Z_i) \quad \forall i, \tag{2}$$

where $\hat{\mathrm{Pa}}(X_i)$ denotes the generated causal parents of $X_i$ (for root nodes the empty set), and each $Z_i$ is independently sampled from $P(Z)$ (e.g. standard Gaussian). We denote the full sequential generator by $G(Z) = [G_1(Z_1), ..., G_d(\cdot, Z_d)]$.

Subsequently, the synthetic sample $\hat{\mathbf{x}}$ is passed to a discriminator $D : \mathbb{R}^d \to \mathbb{R}$, which is trained to distinguish the generated samples from original samples. A typical minimax objective is employed for creating generated samples that confuse the discriminator most:

$$\max_{\{G_i\}_{i=1}^d} \min_D \mathbb{E}[\log D(G(Z)) + \log(1 - D(X)], \tag{3}$$

with $X$ sampled from the original data. We optimize the discriminator and generator iteratively and add a regularization loss to both networks. Network parameters are updated using gradient descent.

If we assume $P_X(X)$ is compatible with graph $\mathcal{G}$, we can show that the sequential generator has the same theoretical convergence guarantees as standard GANs [20]:

**Theorem 2.** *(Convergence guarantee) Assuming the following three conditions hold:*

  (i) *data generating distribution $P_X$ is Markov compatible with a known DAG $\mathcal{G}$;*

 (ii) *generator $G$ and discriminator $D$ have enough capacity; and*

(iii) *in every training step the discriminator is trained to optimality given fixed $G$, and $G$ is subsequently updated as to maximize the discriminator loss (Eq. 3);*

*then generator distribution $P_G$ converges to true data distribution $P_X$*

*Proof.* See Appendix B ☐

Condition (i), compatibility with $\mathcal{G}$, is a weaker assumption than assuming perfect causal knowledge. For example, suppose the Markov equivalence class of the true underlying DAG has been determined through causal discovery. In that case, any graph $\mathcal{G}$ in the equivalence class is compatible with the data and can thus be used for synthetic data generation. However, we note that debiasing can require the correct directionality for some definitions of fairness, see Discussion.

**Remark.** The causal GAN we propose, DECAF, is simple and extendable to other generative methods, e.g., VAEs. Furthermore, from the post-processing theorem [35] it follows that DECAF can be directly used for generating *private* synthetic data by replacing the standard discriminator by a differentially private discriminator [2, 36].

## 5.2 Inference-time Debiasing

The training phase yields conditional generators $\{G_i\}_{i=1}^d$, which can be sequentially applied to generate data with the same output distribution as the original data (proof in Appendix B). The causal model allows us to go one step further: when the original data has characteristics that we do not want to propagate to the synthetic data (e.g., gender bias), individual generators can be modified to remove these characteristics. Given the generator's graph $\mathcal{G} = (X, E)$, fairness is achieved by removing edges such that the fairness criteria are met, see Section 4. Let $\tilde{E} \in E$ be the set of edges to remove for satisfying the required fairness definition. For CF, FTU and DP,[5] the sets $\tilde{E}$ are given by Corollaries 1, 2 and 3, respectively.

Removing an edge constitutes to what we call a "surrogate" $do$-operation [27] on the conditional distribution. For example, suppose we only want to remove $(i \rightarrow j)$. For a given sample, $X_i$ is generated normally (Eq. 2), but $X_j$ is generated using the modified:

$$\hat{X}_j^{do(X_i)=\tilde{x}_{ij}} = G_j(..., X_i = \tilde{x}_{ij}), \tag{4}$$

where $X_i = \tilde{x}_{ij}$ is the surrogate parent assignment. Value $\hat{X}_j^{do(X_i)}$ can be interpreted as the counterfactual value of $\hat{X}_j$, had $X_i$ been equal to $\tilde{x}_{ij}$ (see also [15]).

Choosing the value of surrogate variable $\tilde{x}_{ij}$ requires background knowledge of the task and bias at hand. For example, surrogate variable $\tilde{x}_{ij}$ can be sampled independently from a distribution for each synthetic sample (e.g., the marginal $P(X_i)$), be set to a fixed value for all samples in the synthetic data (e.g., if $X_i$: gender, always set $\tilde{x}_{ij} = male$ when generating feature $X_j$: job opportunity) or be chosen as to maximize/minimize some feature (e.g. $\tilde{x}_{ij} = \arg\max_x \hat{X}_j^{do(X_i)=x}$). We emphasize that we do not set $X_i = \tilde{x}_{ij}$ in the synthetic sample; $X_i = \tilde{x}_{ij}$ is only used for substitution of the removed dependence. We provide more details in Appendix E.

More generally, we create surrogate variables for all edges we remove, $\{\tilde{x}_{ij} : (i \rightarrow j) \in \tilde{E}\}$. Each sample is sequentially generated by Eq. 4, with a surrogate variable for each removed incoming edge.

**Remark.** Multiple datasets can be created based on different definitions of fairness and/or different downstream prediction targets. Because debiasing happens at inference-time, this does not require retraining the model.

## 6 Experiments

In this section, we validate the performance of DECAF for synthesizing bias-free data based on two datasets: i) real data with existing bias and ii) real data with synthetically injected bias. The aim of the former is to show that we can remove real, existing bias. The latter experiment provides a ground-truth unbiased target distribution, which means we can evaluate the quality of the synthetic dataset with respect to this ground truth. For example, when historically biased data is first debiased, a model trained on the synthetic data will likely create better predictions in contemporary, unbiased/less-biased settings than benchmarks that do not debias first.

In both experiments, the ground-truth DAG is unknown. We use causal discovery to uncover the underlying DAG and show empirically that the performance is still good.

**Benchmarks.** We compare DECAF against the following benchmark generative methods: a GAN, a Wasserstein GAN with gradient penalty (WGAN-GP) [21] and FairGAN [17]. FairGAN is the only benchmark designed to generate synthetic fair data,[6] whereas GAN and WGAN-GP only aim to match the original data's distribution, regardless of inherent underlying bias. For these benchmarks, fair data can be generated by naively removing the protected variable – we refer to these methods with the PR (protected removal) suffix and provide more experimental results and insight into PR in Appendix A. We benchmark DECAF debiasing in four ways: i) with *no inference-time debiasing*

---

[5]Just like in Corollaries 1 and 3, we assume the downstream evaluation distribution is the same as the biased training data distribution: a predictor trained on the synthetic debiased data, is required to give fair predictions in real-life settings with distribution $P_X(X)$.

[6]The works of [11, 16] are not applicable here, as these methods are constrained to discrete data.

Table 2: Bias removal experiment on the Adult dataset [40]. The full table with protected attribute removal can be found in Appendix A.

| | Data Quality | | | Fairness | |
|---|---|---|---|---|---|
| **Method** | Precision↑ | Recall↑ | AUROC↑ | FTU↓ | DP↓ |
| Original data $\mathcal{D}$ | $0.920 \pm 0.006$ | $0.936 \pm 0.008$ | $0.807 \pm 0.004$ | $0.116 \pm 0.028$ | $0.180 \pm 0.010$ |
| GAN | $0.607 \pm 0.080$ | $0.439 \pm 0.037$ | $0.567 \pm 0.132$ | $0.023 \pm 0.010$ | $0.089 \pm 0.008$ |
| WGAN-GP | $0.683 \pm 0.015$ | $0.914 \pm 0.005$ | $0.798 \pm 0.009$ | $0.120 \pm 0.014$ | $0.189 \pm 0.024$ |
| FairGAN | $0.681 \pm 0.023$ | $0.814 \pm 0.079$ | $0.766 \pm 0.029$ | $0.009 \pm 0.002$ | $0.097 \pm 0.018$ |
| DECAF-ND | $0.780 \pm 0.023$ | $0.920 \pm 0.045$ | $0.781 \pm 0.007$ | $0.152 \pm 0.013$ | $0.198 \pm 0.013$ |
| DECAF-FTU | $0.763 \pm 0.033$ | $0.925 \pm 0.040$ | $0.765 \pm 0.010$ | $0.004 \pm 0.004$ | $0.054 \pm 0.005$ |
| DECAF-CF | $0.743 \pm 0.022$ | $0.875 \pm 0.038$ | $0.769 \pm 0.004$ | $0.003 \pm 0.006$ | $0.039 \pm 0.011$ |
| DECAF-DP | $0.781 \pm 0.018$ | $0.881 \pm 0.050$ | $0.672 \pm 0.014$ | $0.001 \pm 0.002$ | $0.001 \pm 0.001$ |

(DECAF-ND), ii) under FTU (DECAF-FTU), iii) under CF (DECAF-CF) and iv) under DP fairness (DECAF-DP). We provide DECAF[7] implementation details in Appendix D.1.

**Evaluation criteria.** We evaluate DECAF using the following metrics:
- **Data quality** is assessed using metrics of precision and recall [37, 38, 39]. Additionally, we evaluate all methods in terms of AUROC of predicting the target variable using a downstream classifier (MLP in these experiments) trained on synthetic data.
- **FTU** is measured by calculating the difference between the predictions of a downstream classifier for setting $A$ to 1 and 0, respectively, such that $|P_{A=0}(\hat{Y}|X) - P_{A=1}(\hat{Y}|X)|$, while keeping all other features the same. This difference measures the direct influence of $A$ on the prediction.
- **DP** is measured in terms of the *Total Variation* [15]: the difference between the predictions of a downstream classifier in terms of positive to negative ratio between the different classes of protected variable $A$, i.e., $|P(\hat{Y}|A = 0) - P(\hat{Y}|A = 1)|$.

## 6.1 Debiasing Census Data

In this experiment, we are given a biased dataset $\mathcal{D} \sim P(X)$ and wish to create a synthetic (and debiased) dataset $\mathcal{D}'$, with which a downstream classifier can be trained and subsequently be rolled out in a setting with distribution $P(X)$. We experiment on the Adult dataset [40], with known bias between `gender` and `income` [10, 11]. The Adult dataset contains over 65,000 samples and has 11 attributes, such as `age`, `education`, `gender`, `income`, among others. Following [11], we treat `gender` as the protected variable and use `income` as the binary target variable representing whether a person earns over \$50K or not. For DAG $\mathcal{G}$, we use the graph discovered and presented by Zhang et al. [11]. In Appendix D.2, we specify edge removals for DECAF-DP, DECAF-CF, and DECAF-FTU.

Synthetic data is generated using each benchmark method, after which a separate MLP is trained on each dataset for computing the metrics; see Appendix D.2 for details. We repeat this experiment 10 times for each benchmark method and report the average in Table 2. As shown, DECAF-ND (no debiasing) performs amongst the best methods in terms of data utility. Because the data utility in this experiment is measured with respect to the original (biased) dataset, we see that the methods DECAF-FTU, DECAF-CF, and DECAF-DP score lower than DECAF-ND because these methods distort the distribution – with DECAF-DP distorting the label's conditional distribution most and thus scoring worst in terms of AUROC. Note also that a downstream user who is only focused on performance would choose the synthetic data from WGAN-GP or DECAF-ND, which are also the most biased methods. Thus, we see that there is a trade-off between fairness and data utility when the evaluation distribution $P(X)$ is the original biased data.

## 6.2 Fair Credit Approval

In this experiment, direct bias, which was not previously present, is synthetically injected into a dataset $\mathcal{D}$ resulting in a biased dataset $\tilde{\mathcal{D}}$. We show how DECAF can remove the injected bias, resulting in dataset $\mathcal{D}'$ that can be used to train a downstream classifier. This is a relevant scenario if the training data $\tilde{D}$ does not follow real-world distribution $P(X)$, but instead a biased distribution $\tilde{P}(X)$ (due to, e.g., historical bias). In this case, we want downstream models trained on synthetic data $\mathcal{D}'$ to perform well on the real-world data $\mathcal{D}$ instead of $\tilde{\mathcal{D}}$. We show that DECAF is successful at removing the bias and how this results in higher data utility than benchmarks methods trained on $\tilde{D}$.

---

[7] `PyTorch Lightning` source code at https://github.com/vanderschaarlab/DECAF.

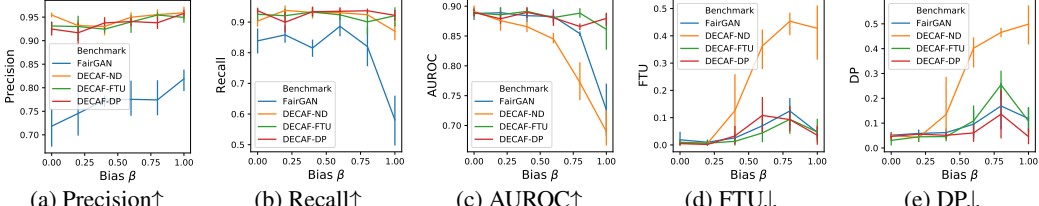

| (a) Precision↑ | (b) Recall↑ | (c) AUROC↑ | (d) FTU↓ | (e) DP↓ |

Figure 3: Plot of precision **(a)**, recall **(b)**, AUROC **(c)**, FTU **(d)**, and DP **(e)** over bias strength $\beta$. FairGAN performs similarly in terms of DP and FTU, but DECAF-FTU and DECAF-DP have significantly better data quality as well as down stream prediction capability (AUROC).

We use the Credit Approval dataset from [40], with graph $\mathcal{G}$ as discovered by the causal discovery algorithm FGES [41] using Tetrad [42] (details in Appendix D.3). We inject direct bias by decreasing the probability that a sample will have their credit approved based on the chosen $A$.[8] The `credit_approval` for this population was synthetically denied (set to 0) with some bias probability $\beta$, adding a directed edge between label and protected attribute.

In Figure 3, we show the results of running our experiment 10 times over various bias probabilities $\beta$. We benchmark against FairGAN, as it is the only benchmark designed for synthetic debiased data. Note that in this case, the causal DAG has only one indirect biased edge between the protected variable (see Appendix D), and thus DECAF-DP and DECAF-CF remove the same edges and are the same for this experiment. The plots show that DECAF-FTU and DECAF-DP have similar performance to FairGAN in terms of debiasing; however, all of the DECAF-* methods have significantly better data quality metrics: precision, recall, and AUROC. DECAF-DP is one of the best performers across all 5 of the evaluation metrics and has better DP performance under higher bias. As expected, DECAF-ND (no debiasing) has the same data quality performance in terms of precision and recall as DECAF-FTU and DECAF-DP and has diminishing performance in terms of downstream AUROC, FTU, and DP as bias strength increases. See Appendix D for other benchmarks, and the same experiment under hidden confounding in Appendix G.

## 7 Discussion

We have proposed DECAF, a causally-aware GAN that generates fair synthetic data. DECAF's sequential generation provides a natural way of removing these edges, with the advantage that the conditional generation of other features is left unaltered. We demonstrated on real datasets that the DECAF framework is both versatile and compatible with several popular definitions of fairness. Lastly, we provided theoretical guarantees on the generator's convergence and fairness of downstream models. We next discuss limitations as well as applications and opportunities for future work.

**Definitions.** DECAF achieves fairness by removing edges between features, as we have shown for the popular FTU and DP definitions. Other independence-based [30] fairness definitions can be achieved by DECAF too, as we show in Appendix C. Just like related debiasing works [10, 11, 16, 17], DECAF is not compatible with fairness definitions based on separation or sufficiency [30], as these definitions depend on the downstream model more explicitly (e.g. Equality of Opportunity [12]). More on this in Appendix C.

**Incorrect DAG specification.** Our method relies on the provision of causal structure in the form of a DAG for i) deciding the sequential order of feature generation and ii) deciding which edges to remove to achieve fairness. This graph need not be known a priori and can be discovered instead. If discovered, the DAG needs not equal the true DAG for many definitions of fairness, including FTU and DP, but only some (in)dependence statements are required to be correct (see Proposition 1). This is shown in the Experiments, where the DAG was discovered with the PC algorithm [47] and TETRAD [42]. Furthermore, in Appendix B we prove that the causal generator converges to the right distribution for any graph that is Markov compatible with the data. We reiterate, however, that knowing (part of) the true graph is still helpful because i) it often leads to simpler functions $\{f_i\}_{i=1}^{d}$ to approximate,[9] and ii) some causal fairness definitions do require correct directionality—see Appendix

---

[8]We let $A$ equal (anonymized) `ethnicity` [43, 44, 45, 46], with randomly chosen $A = 4$ as the disadvantaged population.

[9]Specifically, this is the case if modeling the causal direction is simpler than modeling the anti-causal direction. For many classes of models this is true when algorithmic independence holds, see [34].

C. In Appendix F, we include an ablation study on how errors in the DAG specification affect data quality and downstream fairness.

**Causal sufficiency.** We have focused on just one type of graph: causally-sufficient directed graphs. Extending this to undirected or mixed graphs is possible as long as the generation order reflects a valid factorization of the observed distribution. This includes settings with hidden confounders. We note that for some definitions of bias, e.g., counterfactual bias, directionality is essential and hidden confounders would need to be corrected for (which is not generally possible).

**Time-series.** We have focused on the tabular domain. The method can be extended to other domains with causal interaction between features, e.g., time-series. Application to image data is non-trivial, partly because, in this instance, the protected attribute (e.g., skin color) does not correspond to a single observed feature. DECAF might be extended to this setting in the future by first constructing a graph in a disentangled latent space (e.g., [24, 25]).

**Social implications.** Fairness is task and context-dependent, requiring careful public debate. With that being said, DECAF empowers data issuers to take responsibility for downstream model fairness. We hope that this progresses the ubiquity of fairness in machine learning.

## Acknowledgements

We would like to thank the reviewers for their time and valuable feedback. This research was funded by the *Office of Naval Research* and the *WD Armstrong Trust*.

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
