# Supplementary Material for "DECAF: Generating Fair Synthetic Data Using Causally-Aware Generative Networks"

**Boris van Breugel***
University of Cambridge
bv292@cam.ac.uk

**Trent Kyono***
University of California, Los Angeles
tmkyono@ucla.edu

**Jeroen Berrevoets**
University of Cambridge
jb2384@cam.ac.uk

**Mihaela van der Schaar**
University of Cambridge
University of California, Los Angeles
The Alan Turing Institute
mv472@cam.ac.uk

## A    Protected variable removal

A trivial method for satisfying FTU fairness, is to remove the protected attribute from downstream learners. We first provide a motivating example explaining why this is sub-optimal. We then follow this with an experiment on the Adult dataset.

### A.1    Example

Defining fairness is task and data dependent. For example, let us assume two datasets are generated by the graphical models in Figure 1. Data generated by the top graph is considered fair: *Education* affects past experience (*Resume*), which together affect future job prospects (*Job*). The bottom graph is a historical example of unfairness: even if there would be no bias between *Loan* and *Race*, *redlining* (i.e. the practice of refusing a loan to people living in certain areas) would discriminate indirectly based on race [1, 2, 3, 4]. Human knowledge is thus essential for defining fairness correctly, and making sure (e.g., historical) bias is not propagated by the models we deploy. This example also shows why simply removing or not measuring a sensitive attribute does not suffice: not only does this ignore indirect bias, but hiding the protected attribute leads to an (additional) correlation between *Postcode* and *Loan* due to confounding. A smart debiasing method is required that can distinguish fair from unfair relations.

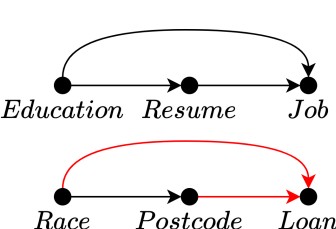

Figure 1: Human knowledge is essential for defining fairness.

### A.2    Experiment

As explained in the previous example, simply removing the protected attribute is a naive and sub-optimal solution to FTU fairness due to confounding. Let us test this experimentally. We use the same experimental setup described in Section 6 for the Adult dataset, but we include additional metrics for protected attribute removal. We denote protected attribute removal by the *-PR suffix. In Table 1, we observe that naively removing the protected attribute only ensures FTU fairness, as shown by: GAN-PR, WGAN-GP-PR, and DECAF-PR. Furthermore, we observe that synthetic data quality

---

*Equal contribution

35th Conference on Neural Information Processing Systems (NeurIPS 2021).

diminishes as well for WGAN-GP-PR and DECAF-PR across precision, recall, and AUROC. For GAN-PR we see a slight improvement in data quality over GAN, however this improvement is very minimal in comparison to DECAF.

Table 1: Full table of bias removal experiment on Adult dataset [5] including protected removal (PR) metrics. For methods *-PR, we remove the protected attribute from the dataset before synthesizing data. ‡Note that the FTU values for the *-PR values will be zero since they are removed from the data generation method.

| Method | Data Quality | | | Fairness | |
|---|---|---|---|---|---|
| | Precision↑ | Recall↑ | AUROC↑ | FTU↓ | DP↓ |
| Original data $\mathcal{D}$ | $0.920 \pm 0.006$ | $0.936 \pm 0.008$ | $0.807 \pm 0.004$ | $0.116 \pm 0.028$ | $0.180 \pm 0.010$ |
| GAN | $0.607 \pm 0.080$ | $0.439 \pm 0.037$ | $0.567 \pm 0.132$ | $0.023 \pm 0.010$ | $0.089 \pm 0.008$ |
| WGAN-GP | $0.683 \pm 0.015$ | $0.914 \pm 0.005$ | $0.798 \pm 0.009$ | $0.120 \pm 0.014$ | $0.189 \pm 0.024$ |
| FairGAN | $0.681 \pm 0.023$ | $0.814 \pm 0.079$ | $0.766 \pm 0.029$ | $0.009 \pm 0.002$ | $0.097 \pm 0.018$ |
| GAN-PR | $0.632 \pm 0.077$ | $0.509 \pm 0.110$ | $0.612 \pm 0.106$ | ‡$0.0 \pm 0.0$ | $0.120 \pm 0.012$ |
| WGAN-GP-PR | $0.640 \pm 0.019$ | $0.848 \pm 0.028$ | $0.739 \pm 0.034$ | ‡$0.0 \pm 0.0$ | $0.078 \pm 0.014$ |
| DECAF-PR | $0.717 \pm 0.021$ | $0.839 \pm 0.033$ | $0.769 \pm 0.020$ | ‡$0.0 \pm 0.0$ | $0.044 \pm 0.013$ |
| DECAF-ND | $0.780 \pm 0.023$ | $0.920 \pm 0.045$ | $0.781 \pm 0.007$ | $0.152 \pm 0.013$ | $0.198 \pm 0.013$ |
| DECAF-FTU | $0.763 \pm 0.033$ | $0.925 \pm 0.040$ | $0.765 \pm 0.010$ | $0.004 \pm 0.004$ | $0.054 \pm 0.005$ |
| DECAF-CF | $0.743 \pm 0.022$ | $0.875 \pm 0.038$ | $0.769 \pm 0.004$ | $0.003 \pm 0.006$ | $0.039 \pm 0.011$ |
| DECAF-DP | $0.781 \pm 0.018$ | $0.881 \pm 0.050$ | $0.672 \pm 0.014$ | $0.001 \pm 0.002$ | $0.001 \pm 0.001$ |

## B Convergence guarantees DECAF GAN

Assuming the correct underlying data generating DAG is known, well-known theoretical results for GANs transfer to DECAF. We highlight the main results. The typical GAN minimax objective (Eq. 3 paper) is optimized by iteratively updating the discriminator and generator, with respective losses:

$$\mathcal{L}_D(\hat{X}, X) = \log D(\hat{X}) + \log(1 - D(X)) \tag{1}$$

$$\mathcal{L}_G(\hat{X}) = -\log D(\hat{X}) \tag{2}$$

First, we reiterate the following theorem from [6]. Let $P_G$ and $P_X$ denote generator and original data distributions, respectively.

**Theorem 1.** *Given fixed optimal discriminator $D^*$, the global minimum of the generator loss (Eq. 2) is achieved if and only if $P_G = P_X$.*

*Proof.* Noting that we have made no changes to the GAN discriminator, we refer to Theorem 1 of [6]. □

**Theorem 2.** *(Convergence guarantee) Assuming the following three conditions hold:*

*(i) data generating distribution $P_X$ is Markov compatible with a known DAG $\mathcal{G} = (V, E)$;*

*(ii) generator $G$ and discriminator $D$ have enough capacity; and*

*(iii) in every training step the discriminator is trained to optimality given fixed $G$, and $G$ is subsequently updated as to maximise the discriminator loss (Eq. 3 paper);*

*then generator distribution $P_G$ converges to true data distribution $P_X$*

*Proof.* This is the direct result of the construction of generator $G$ and follows a similar argument as Proposition 2 of [6]. Note that by the definition of compatibility of $P_X$ and $\mathcal{G} = (V, E)$, we can write:

$$P_X(X) = \prod_{X_i \in V} P(X_i | \{X_j : (X_j \to X_i) \in E\})$$

Given each $G_i$ (see Eq. 2 paper) has enough capacity, $G$ can thus express the full distribution $P_X(X)$. By convexity of the loss functions and the existence of a unique global optimum (Theorem 1), gradient descent is theoretically guaranteed to converge, $P_G \to P_X$ [6]. □

Note that for condition (i) of Theorem 2 to be valid, we do not require that graph $\mathcal{G}$ equals the true underlying DAG of the data generating distribution $P_X$; $P_G$ is only required to disentangle into the causal factors implied by $\mathcal{G}$. This is highly beneficial, as it enables generation of perfect synthetic data without perfect causal knowledge. For example, if the Markov equivalence class of the true underlying DAG has been determined through causal discovery, any graph $\mathcal{G}$ in the equivalence class satisfies condition (i) of Theorem 2.

**Remarks** The convergence guarantees do not necessarily hold in practice. First, finite data means there there is no guarantee the algorithm converges to the true underlying data distribution instead of, for example, the observed empirical data distribution. Second, in practice each generator $G_i$ will have limited capacity and $P(X_i \mid \mathrm{Pa}(X_i))$ might not lie in its support. On a more positive note, these limitations are not specific for DECAF and generally GANs have done well in the past. Additionally, our method is directly extendable to the more stable WGAN-GP [7] and other generative models.

## C  Compatibility different fairness definitions

**Related definitions** In the paper we have discussed FTU, DP and CF, which are independence-based definitions and do not take directionality explicitly into account when defining fairness. Some authors use similar definitions, but instead of looking at (conditional) independencies of $A$ and $Y$, they consider (blocked) directed paths from protected attribute $A$ to $Y$. These definitions are compatible with DECAF, but mean less edges need to be removed. See Table 2 and Figure 2. Zhang et al. [8] consider direct and indirect discrimination, which can be understood as the "directed path" equivalents of FTU and DP.[2] Assuming faithfulness and not allowing any discrimination—i.e. $\tau = 0$ in [8]—direct and indirect discrimination prohibit the existence of edge $A \to Y$ and directed path $A$ to $Y$, respectively. Zhang and Bareinboim [9] disentangle the total effect of $A$ on $Y$ into direct, indirect and spurious relations. This leads to the same definition for direct discrimination as [8], but a different definition of indirect discrimination as it *does* allow for direct influence of $A$ on $Y$. A very similar definition, coined counterfactual fairness, is proposed by [10]. Kilbertus et al. [11] introduce *unresolved discrimination* (UD) as the path-equivalent version of conditional fairness. They define *proxy* discrimination as well, which can be considered the dual of UD [11].

**Incompatible definitions** Some definitions are not compatible with fair synthetic data generation because they rely on the final prediction, e.g. equality of opportunity [12] and calibration (e.g. see [13]). As a consequence, DECAF cannot be used for these. Furthermore, we note that all our fairness definitions are binary: a distribution is fair or unfair. In practice some level of unfairness might be tolerated. For example, the US Supreme Court's 80% rule [14] essentially states that a prediction has disparate impact if for disadvantaged group $A = 1$ and positive outcome $\hat{Y} = 1$, $\frac{P(\hat{Y}=1|A=1)}{P(\hat{Y}=1|A=0)} < 0.8$ [15]. Some authors (e.g. Feldman et al. [15]) have explored this continuous definition, but because it requires quantification of path-specific effects work is limited by a linearity assumption. Extending this to nonlinear path-specific effects is an interesting direction for future work, with great relevance for real-life applications.

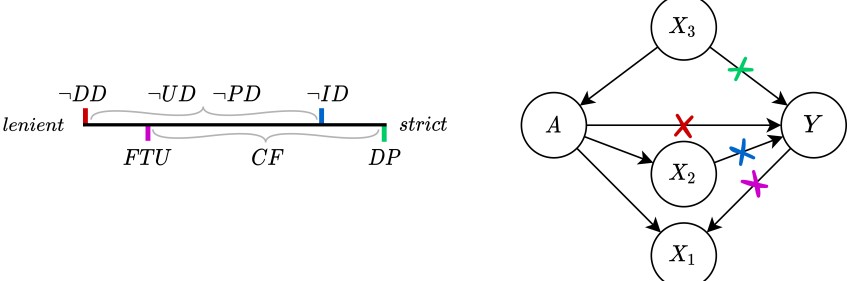

Figure 2: (Left) Typical strictness of different definitions. Note that the strictness of CF, $\neg$ UD and $\neg PD$ depends on the choice of explanatory variables/proxies. (Right) Example showing different definitions and required edge removals. $\neg$ DD: ✗; FTU: ✗✗; $\neg$ ID: ✗✗; DP: ✗✗✗✗. Note that for FTU, $A \to X_1$ could have been removed instead of $Y \to X_1$.

---

[2]Note: the legal definitions of direct and indirect discrimination are in fact defined as FTU and DP.

Table 2: Different definitions of fairness that are compatible with DECAF and which edges need removal when evaluation distribution $P(X) = P_X(X)$. The first three definitions are non-causal, the others only prohibit causal paths. $A, Y, P, R$ denote respectively the protected attribute, label, proxy variables and explanatory variables. Let $\pi_{A \to Y}$ denote a directed path from $A$ to $Y$ that ends with $B \to Y$ for some $B$.

| Definition | Edges to remove |
|---|---|
| Demographic Parity (DP) [16] | $B \leftrightarrow Y : \forall B \in Bl_{\mathcal{G}'}(Y)$ with $A \not\perp\!\!\!\perp B$ |
| Conditional Fairness (CF) | $B \leftrightarrow Y : \forall B \in Bl_{\mathcal{G}'}(Y)$ with $A \not\perp\!\!\!\perp B|R$ |
| Fairness through Unawareness (FTU) | $A \leftrightarrow Y$ and $(A \to C$ or $Y \to C : \forall C$ with $A \to C \leftarrow Y)$ |
| No Indirect Discrim. ($\neg$ ID) [8] | $B \to Y$ if there exists $\pi_{A \to Y}$ |
| No Proxy Discrim. ($\neg$PD) [11] | $B \to Y$ if there exists $\pi_{A \to Y}$ that is blocked by $P$ |
| No Unresolved Discrim. ($\neg$UD) [11] | $B \to Y$ if there exists $\pi_{A \to Y}$ that is not blocked by $R$ |
| No Direct Discrim. ($\neg$ DD) [8, 9] | $A \to Y$ |

# D  Additional Details and Results

## D.1  Implementation details.

We instantiate the generator of DECAF with $d$ sub-networks with shared hidden layers. Both the generator and discriminator are constructed having 2 hidden layers with $2d$ neurons and initialized with random uniform weights. Each benchmark is initialized with the same random weights and published hyperparameters. For preprocessing, all continuous variables are standardized. We use the Adam optimizer with a learning rate of $0.001$ for up to 50 epochs. We update the generator once for every 10 discriminator updates. We implement DECAF using PyTorch Lightning[3].

**Computational hardware.**  All models were trained on an Ubuntu 18.04 OS with 64GB of RAM (Intel Core i7-6850K CPU @ 3.60GHz) and 2 NVidia 1080 Ti GPUs.

**Scalability** Due to the sequential feature generation, DECAF's run time scales linearly with the number of variables. In practice—for the larger Communities and Crime dataset—this comes down to an average training time of just about 35s per epoch when run on a machine with hexacore Intel i7-6850K CPU. Practical improvements can be made to speed this up further: when the graph is sparse one can parallelize calculations and often one can cluster (some) variables and model clusters together using a single generator network.

**Generating discrete variables**  In both datasets the only non-binary discrete variable is the protected attribute, which for simplicity we have binarised (discriminated vs non-discriminated). All variables are generated in the same way, but binary variables are rounded off after generation.

Table 3: Overview datasets

|  | Credit | Census | Communities |
|---|---|---|---|
| Number of features | 15 | 10 | 128 |
| - Continuous | 3 | 4 | 120 |
| - Discrete | 12 | 6 | 8 |
| Target type | Binary | Binary | Binary |
| Number of samples | 379 | 32,561 | 1994 |
| Number of discovered edges | 40 | 22 | 1288 |

## D.2  Census Dataset Details

DECAF supports both FTU and DP debiasing, i.e. respectively direct and indirect discrimination removal. We use the DAG from [8, 15] as shown in Figure 3. FTU is achieved

---

[3]Source code is available at https://github.com/vanderschaarlab/DECAF

by removing the directed edge between between `sex` and `income` (see Corollary 3), DP is achieved by removing[4] all incoming edges into the target variable that have the protected variable as an ancestor (Corollary 2)- these include edges between the target variable `income` and each of `occupation`, `hours_per_week`, `occupation`, `workclass`, `education`, `relationship`, `marital_status`, and `sex`. DP fairness is overly strict, so to satisfy CF fairness, we allow the variables `occupation`, `hours_per_week`, `workclass`, and `education` while removing the edges from `sex`, `marital_status`, and `relationship`.

We generate synthetic data from the ground truth dataset using each benchmark generator. We randomly hold out a sample of 2000 samples as a test set. We train an MLP using default `scikit-learn` hyperparameters on the generated dataset to use as our downstream classifier. We use a hidden layer with 100 neurons and ReLU activation functions. For the output layer we use a softmax activation and binary cross entropy loss. We use Adam as the optimizer with a learning rate of 0.001.

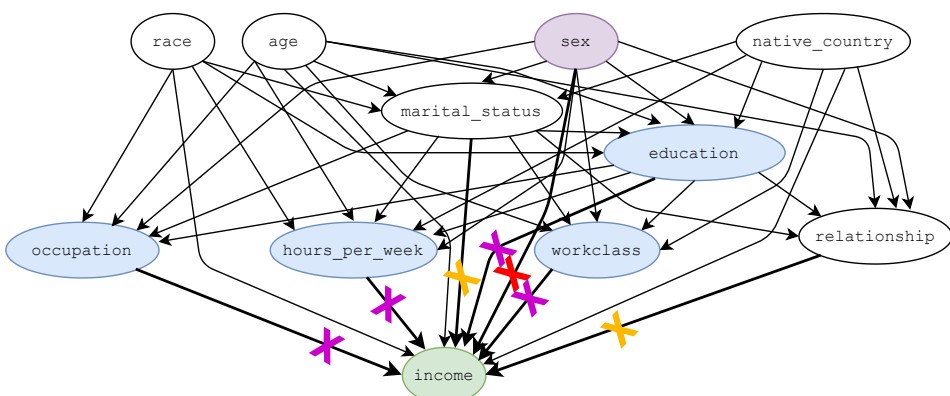

Figure 3: Adult dataset DAG from [8, 15]. The target variable is in green, the protected attribute in purple, and the allowed CF variables in blue. *FTU is achieved by removing: ✗; DP: ✗✗✗; CF: ✗✗.* In this particular instance, we follow [17], and remove gender discrimination. However, our method generalizes to removing the highly problematic variable `race` to `income`.

### D.3 Fair Credit Details

We use the Credit Approval dataset from [5] as our GT dataset. We synthetically add bias by decreasing the probability that a sample will be have their credit approved based on the chosen $A$. We induce bias by choosing $A$ to be `ethnicity` [1, 2, 3, 4], with a discriminated population having a value of $4$[5]. The `credit_approval` for this population was synthetically denied (set to 0) with some bias probability $\beta$ – see Section 6.2 for more details.

The causal DAG used in this experiment is shown in Figure 4. This DAG was found using the Fast Greedy Equivalence Search (FGES) [18] with the `pycausal` library [19]. We provide the prior knowledge that `age` and `ethnicity` are root nodes to the FGES algorithm.

We train an MLP using default `scikit-learn` hyperparameters on the generated dataset to use as our downstream classifier. We use a hidden layer with 100 neurons and ReLU activation functions. For the output layer we use a softmax activation and binary cross entropy loss. We use Adam as the optimizer with a learning rate of 0.001.

In Table 4, we show the results of running this experiment 10 times over our biased dataset. Note that our method was able to generate synthetic examples that had the highest AUROC (demonstrating FTU fairness). Table 4 shows that our method can perform debiasing without performance hits to the synthetic data metrics – i.e., there are no significant difference (outside of a standard deviation) between the top methods.

---

[4]Specifically, we focus on the scenario of $P(X)$ being the original biased data distribution; we want a model trained on synthetic data $\mathcal{D} \sim P'(X)$ to be DP-fair when evaluated on $P(X)$, see remark Section 4.2.

[5]Note that the values have been anonymized in this dataset.

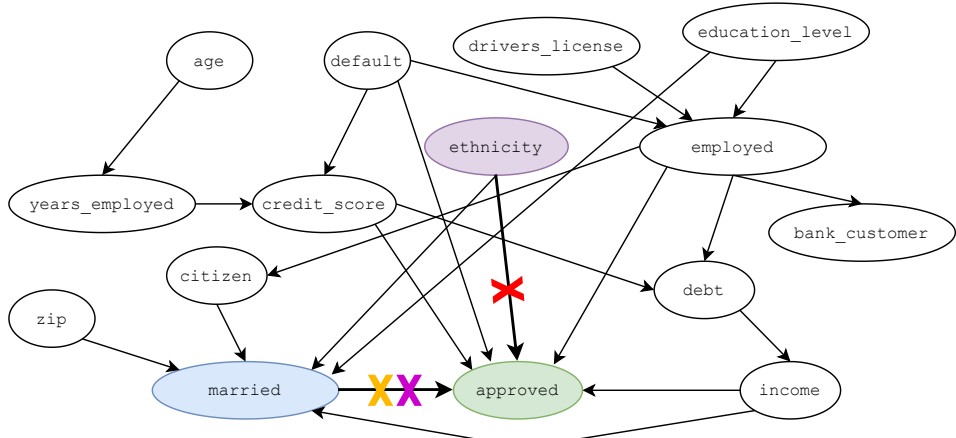

Figure 4: Credit Approval DAG discovered using FGES [18] and `Tetrad` [19]. The target variable is in green, the protected attribute in purple, and the allowed CF variables in blue. *FTU is achieved by removing: ✗; DP: ✗✗✗; CF: ✗✗. Also, note that in this case CF fairness and DP fairness are the same.*

Table 4: Bias removal experiment on Credit Approval dataset. Here we train an MLP on the listed dataset, and report the testing AUROC for credit approval prediction on the ground truth (GT) dataset for the biased population. Methods denoted *-PR represent modifications to the dataset by dropping the protected variable (PR). Note that there the FTU is zero for *-PR methods since the protected variable, P, has been removed.

| Method | Data Quality | | | Fairness | |
| --- | --- | --- | --- | --- | --- |
| | Precision↑ | Recall↑ | AUROC↑ | DP↓ | FTU↓ |
| GAN | $0.921 \pm 0.036$ | $0.335 \pm 0.029$ | $0.743 \pm 0.047$ | $0.405 \pm 0.077$ | $0.194 \pm 0.058$ |
| WGAN | $0.970 \pm 0.007$ | $0.804 \pm 0.057$ | $0.698 \pm 0.009$ | $0.520 \pm 0.036$ | $0.461 \pm 0.029$ |
| ADSGAN | $0.963 \pm 0.009$ | $0.841 \pm 0.052$ | $0.708 \pm 0.009$ | $0.506 \pm 0.013$ | $0.429 \pm 0.059$ |
| GAN-PR | $0.794 \pm 0.117$ | $0.368 \pm 0.080$ | $0.727 \pm 0.047$ | $0.203 \pm 0.196$ | $0.0 \pm 0.0$ |
| WGAN-PR | $0.941 \pm 0.004$ | $0.880 \pm 0.017$ | $0.814 \pm 0.019$ | $0.406 \pm 0.022$ | $0.0 \pm 0.0$ |
| ADSGAN-PR | $0.945 \pm 0.008$ | $0.880 \pm 0.019$ | $0.827 \pm 0.008$ | $0.413 \pm 0.029$ | $0.0 \pm 0.0$ |
| FairGAN | $0.951 \pm 0.012$ | $0.663 \pm 0.046$ | $0.680 \pm 0.008$ | $0.510 \pm 0.075$ | $0.474 \pm 0.054$ |
| DECAF | $0.954 \pm 0.012$ | $0.601 \pm 0.015$ | $0.713 \pm 0.045$ | $0.511 \pm 0.130$ | $0.432 \pm 0.127$ |
| DECAF-FTU | $0.936 \pm 0.017$ | $0.901 \pm 0.034$ | $0.877 \pm 0.009$ | $0.099 \pm 0.065$ | $0.014 \pm 0.012$ |
| DECAF-DP | $0.940 \pm 0.007$ | $0.922 \pm 0.024$ | $0.875 \pm 0.010$ | $0.011 \pm 0.029$ | $0.015 \pm 0.017$ |

# E  Surrogate variables

Debiasing in DECAF relies on removing edges from a trained model. As highlighted in Section 5.2, we need surrogate variables with which to replace the removed edges (Eq. 4 paper). In this section, we compare two surrogate variable mechanisms. The aim is show i) that debiasing is successful independent of the choice of surrogate variables, and ii) how prior knowledge helps in choosing surrogate variable mechanism, which leads to better data quality.

**Mechanisms** Let $\tilde{X}_{ij}$ denote the surrogate variable used for the removed edge $(i \rightarrow j)$, i.e. the surrogate variable that replaces the influence of $X_i$ on $X_j$. Here, we compare two surrogate mechanisms for this setting:

1. $\tilde{X}_{ij} \sim P(X_i)$, i.e. we sample from the parent's marginal distribution,

2. $\tilde{X}_{ij} = \tilde{x}_{ij}$, where $\tilde{x}_{ij}$ is a fixed value.

Mechanism 1 is straightforward and most applicable when one does not know anything about the bias of a particular edge. By sampling from the marginal, each sample might use a different value of

$\tilde{X}_{ij}$ when generating feature $X_j$, which means the diversity of the generated $X_j$ is retained better compared to mechanism 2. Mechanism 1 for all experiments in Section 6.

On the other hand, mechanism 2 is more suitable when we know explicitly that there is bias for some values of $X_i$, e.g. if $X_i$ is the protected attribute we might know there is a group $A = 0$ that is being discriminated. In this case, sampling $\tilde{X}_{ij}$ from the marginal of $A$ is not desired: even though this means we remove direct bias from $A$ to $Y$, it still means we disadvantage some individuals randomly, i.e. every time we sample $\tilde{x}_{ij} = 0$. We can employ the second mechanism instead, i.e. set $\tilde{x}_{ij} = 1$ for all individuals. This corresponds to treating everyone like they are from the advantaged group.

**Experiments** We repeat the experiment from Section 6.2, in which we insert direct bias from $A$ to $Y$ by denying loans for a disadvantaged group $A = 0$ with probability $\beta$. Our aim is to remove the direct bias from $A$ to $Y$ and we evaluate the synthetic data quality and bias with respect to the original, unbiased dataset. As we will see, in this setting mechanism 2 is more appropriate: we want to treat everyone from group $A = 0$ like they are from group $A = 1$, thereby removing the bias we inserted. Meanwhile, we do not want to change the way we generate data for the advantaged group. More specifically, even though it would not be considered discrimination against a protected group, randomly denying loans to individuals of any group should still be considered unfair.

In Figure 5 we plot the quality metrics and FTU for three generation methods: DECAF-ND (no debiasing), DECAF-FTU1 (DECAF-FTU with surrogate mechanism 1) and DECAF-FTU2 (DECAF-FTU with mechanism 2). We plot three columns; on the left we plot the metrics for all generated data, in the middle we plot the metrics as computed on the discriminated group and on the right for the non-discriminated group.

As we can see in the FTU plots (bottom), both debiasing mechanisms are equally valid for removing the injected bias from $A$ to $Y$. However, the precision metric tells a different story. Mechanism 1 disadvantages individuals randomly whenever it samples $\tilde{x}_{ij} = 0$, but this is not in line with what we want the data to be like (no disadvantage like this at all). As a result, we see that the quality of both the discriminated group goes down. The same result can be observed in the recall and AUROC plot, though the overlapping error bars prohibit strong conclusions.

In a nutshell, these results indicate that for different mechanisms for surrogate variables, data fairness is guaranteed. However, knowledge about the origins of the bias can help increase the data utility.

# F   DAG Sensitivity

In this section, we investigate DECAF under imperfect knowledge. Here, we are curious to understand what happens when our causal knowledge has: 1) has missing edges, 2) has spurious edges, i.e., edges that we assumed falsely, and 3) edges that are reversed in directionality.

We perform this experiment on the credit approval dataset [5], with the known DAG used in the manuscript. Using an identical experimental setup as described in Section 6.2 and a bias of $\beta = 0.8$, we run our experiment 10 times each under random DAG perturbations. Starting with the baseline DAG used in our credit approval experiment, we perform a sensitivity analysis to the following DAG perturbations:

- **Edge removal** is done by randomly edges from the baseline DAG.

- **Edge addition** is done by randomly adding edges that are constrained by the following two criteria: 1) it does not create any cycles, and 2) it does not create any new indirect bias measures. For the second condition, we ensure this by asserting that an edge is not added between the protected attribute `ethnicity` and an ancestor of `approved`. We do this to ensure that the indirect bias is held consistent across each DAG instantiation and experimental run.

- **Edge reversal** is done by randomly reversing edges in the baseline DAG while preserving acyclicity.

Results for this experiment are shown in Figure 6. As expected, we see that edge removal degrades synthetic data quality (precision, recall, and AUROC) as the number of edges removed increases; this is not the case for adding and reversing edges – where stable synthetic data quality is preserved.

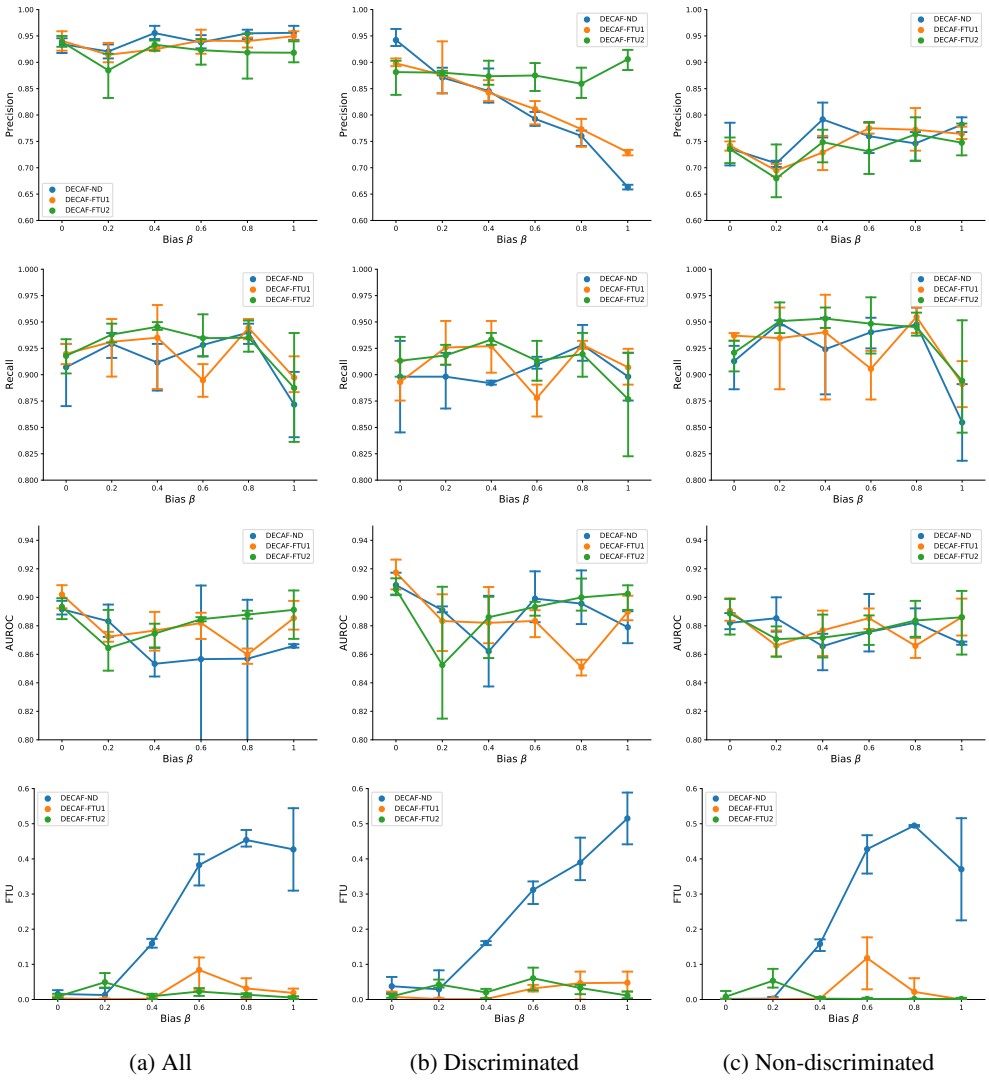

(a) All         (b) Discriminated         (c) Non-discriminated

Figure 5: Plot of precision, recall, AUROC, and FTU over various bias strengths for **(a)** both populations (discriminated and non-discriminated), **(b)** discriminated population, and **(c)** non-discriminated population.

In terms of debiasing, we see that DECAF-FTU and DECAF-ND is still able to debias consistently across all DAG perturbations.

# G   Hidden Confounders

In this section, we examine DECAF under hidden confounders on the Credit Approval dataset. Assuming the DAG in Figure 4, we create a hidden confounder by removing the variable for `education_level` from the dataset and DAG. We then replicate the experimental setup for Section **??** under identical conditions. We present the results in Figure 7.

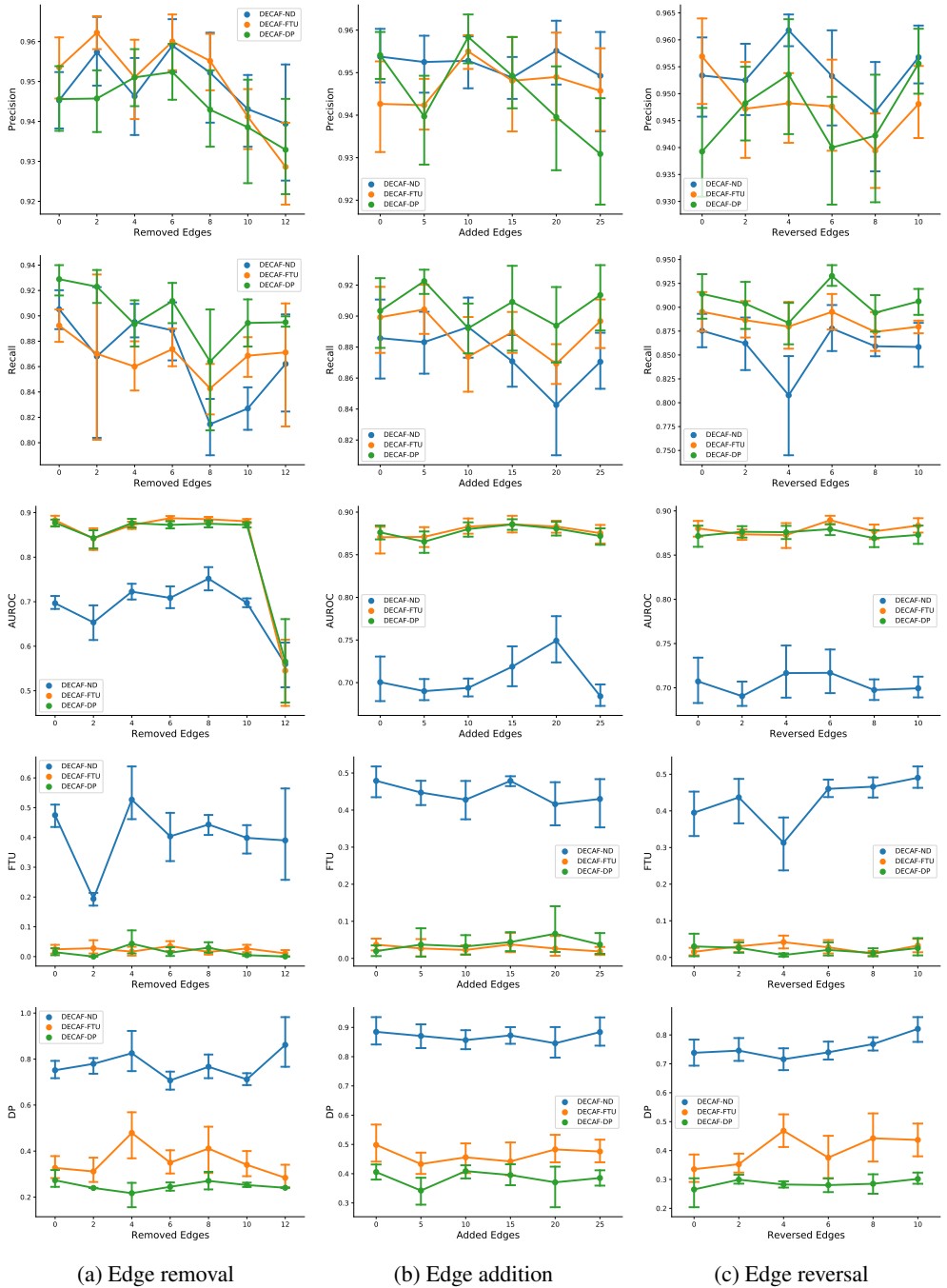

(a) Edge removal      (b) Edge addition      (c) Edge reversal

Figure 6: Plot of precision, recall, AUROC, FTU, and DP over **(a)** edge removal, **(b)** edge addition, and **(c)** edge reversal on the credit approval dataset.

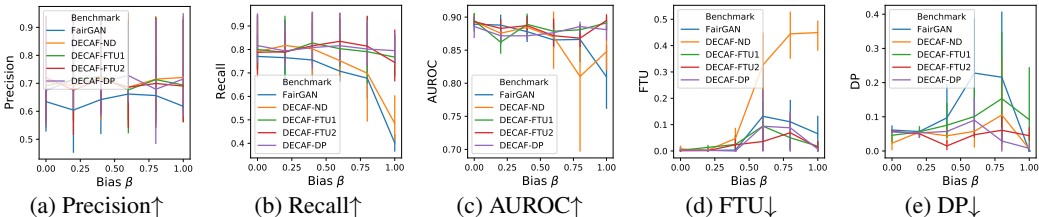

Figure 7: Plot of precision **(a)**, recall **(b)**, AUROC **(c)**, FTU **(d)**, and DP **(e)** over bias strength $\beta$ for experiments with hidden confounding. FairGAN performs similarly in terms of DP and FTU, but DECAF-FTU and DECAF-DP have significantly better data quality as well as down stream prediction capability (AUROC).