# OpenReview forum: "DECAF:  Generating Fair Synthetic Data Using Causally-Aware Generative Networks"
_NeurIPS.cc/2021/Conference — NeurIPS 2021 Poster_

### Official Review · Reviewer_tL8r · 2021-07-02

**Rating:** 6
**Confidence:** 4

**Summary:**

This paper proposes DECAF, a fair data generation method using causal DAG structures and GANs. The DAG is used to remove any causal dependencies that may have a negative effect on the fairness and is general enough to model various fairness measures: FTU, DP, and CF. Then synthetic data is generated according to the DAG, but made realistic using a discriminator that distinguishes the generated samples from the real ones. Experiments show that DECAF has comparable fairness to the state-of-the-art method FairGAN, but significantly outperforms it in terms of precision, recall, and AUROC.

**Ethical Concerns:**

I do not see any ethical concerns.

**Limitations And Societal Impact:**

Yes. Section 7 has a detailed discussion on limitations and societal impact.

**Main Review:**

This paper proposes a novel approach for removing data biases that negatively affect fairness, yet generating realistic data using GAN techniques. There are some concerns on whether the generated data is realistic and whether DECAF should also be compared with weak supervision approaches.

Strong points:

* The unfairness mitigation method is principled and has theoretical guarantees. By removing edges from the DAG, one has a clear idea exactly what bias is being removed.

* The DAG representation is general and can be used to express prominent group fairness measures.

* The experiments show that DECAF outperforms the state-of-the-art method FairGAN among others.

Weak points:

* One concern is whether removing edges is too drastic and may result in unrealistic data. For example, Sec 5.2 mentions that surrogate variables can be sampled independently from a distribution or set to a fixed value. Let us say that we are using the Adult dataset and set all its gender values to male. From a practical point of view, wouldn't the dataset lose too much information? To answer this question, the authors should also show DECAF's discriminator performance because it may actually be able to distinguish generated versus real data better than say FairGAN's real/fake discriminator.

* An alternative approach for fair data generation is to use weak supervision. In particular, the authors should cite and compare DECAF with Choi et al., "Fair Generative Modeling via Weak Supervision", ICML 2020. The idea is to exploit an additional small, unlabeled reference dataset as the supervision signal to generate unbiased data.

* Some notions are used without definitions. In Sec. 3, please define "independent causal mechanism". In Sec. 4.3, define "Markov boundary", "d-separation", and "explanatory feature".

* The experiments section looks a bit thin (accuracy and fairness experiments for two datasets), probably because many experiments are moved to the supplementary. In particular, the Adult dataset DAG is quite interesting and should be more visible. Also, it would be useful to see an ablation study that evaluates DECAF without some of its components (e.g., the discriminator).

* In Sec. 6, please specify how the values of surrogate variables were chosen. There are some details in the supplementary, but it is not clear if the same settings were used in Sec. 6.

=========
The author feedback addresses my concerns. I will keep my rating of 6.

**Time Spent Reviewing:**

5

---

> ### Author Response · Authors · 2021-08-10
> **Response to Reviewer tL8r**
>
> _Thank you for your time and useful feedback. We would like to address each of the five points you raised: (i) drastic information loss due to debiasing, (ii) weak supervision as alternative method for fairness, (iii) definitions from causality, (iv) experimental section, and (v) choice of surrogate variables._
>
> **[i. Drastic edge removal]** For some fairness definitions, many edges are removed from the DAG. Removing so many edges results in a lot of information loss, most notably for Demographic Parity (DP). For these definitions, losing information is *not* due to a fault in our method, but rather due to the strict constraints imposed by these fairness definitions on the distribution (e.g. see Corollary 3 and Theorem 1 for DP). Essentially, information loss is the goal, and any successful synthetic data generator/classifier will *need* to lose information in order to satisfy the fairness definition.
>
> Naturally, as is pointed out by the reviewer, removing edges will change the synthetic distribution from its original counterpart. If the synthetic distribution changes from the real distribution, then so will the sampled data. An original data with many biases, requires many changes which will (and should!) result in different data. Of course, different definitions of fairness yield different definitions of bias. For instance, as noted in Section 4.1 (line 105), conditional fairness might be more appropriate (than DP) if one allows some information to flow from the protected attribute to the label.
>
> Thank you for the suggestion of including the discriminator loss for DECAF and FairGAN as a metric for comparing generation performance. We would like to note that, to the best of our knowledge, discriminator loss cannot be unambiguously compared between GANs, as it is only an indirect measure of the generator performance and too dependent on the hyperparameter setting and current state of the individual discriminators. As a result, it is not a common metric for comparing GAN performance, e.g. the exhaustive and most cited paper on evaluating GAN models [4] leaves it out entirely. On the other hand, a plot of the discriminator loss could be useful for determining the training stability, and otherwise the Wasserstein critic [5] reflects a similar idea (i.e. separating real from fake samples) and is a commonly used metric. If you have any concerns that either might solve, we are happy to include these in the appendix.
>
> **[ii. Weak supervision]** While we agree with you that Choi et al. (2020) is relevant in terms of objective, there is one major limitation to Choi et al. that DECAF does not have; that is the need for an unbiased second dataset. Not only is this a major limitation, it begs the question: “what is meant with an unbiased dataset?”. As we have shown in our paper, one could devise multiple definitions of bias, even more so, this definition may change over time. Using Choi et al., one would have to recollect this alternative dataset (bias-free), each time a new source of bias is identified. No such step is required in DECAF. We agree with R4, however, that Choi et al. is indeed relevant to our work. As such, we will cite and discuss it in our main text in comparison to DECAF.
>
> **[iii. Definitions]** Thank you for highlighting this. We define “Markov boundary” and “d-separation” as in Pearl’s causality framework [1], “independent causal mechanism” as in Proposition 2.1 of [2] and “explanatory feature” as in Section 3.2 of [3]. Assuming the reader is familiar with d-separation, the Markov boundary of variable $X$ in graph $G = (V,E)$ is the minimal markov blanket, i.e. the minimal set of nodes $A\subseteq V\backslash X$ that d-separates $X$ from all other nodes $V\backslash X\cup A$.
>
> An explanatory feature $R$ is a feature that is correlated with $X$ and informative for $Y$, but the information that it passes from $X$ to $Y$ is considered non-discriminatory and often externally provided by law. For example, a bank is allowed to use an applicant’s salary ($R$) when choosing to hire her/him ($Y$), even though salary might be correlated with the protected attribute(s). On the other hand, using an applicant’s postcode is not allowed, so this is not an explanatory feature.
>
> We will replace “independent causal mechanisms” by “distribution conditioned on parents” as we expect this to be clearer to the reader, the other definitions we will include in the main text.
>
> **[iv. Experiments]** Prioritizing theory and method, unfortunately we have had to move some of the experiments to the appendix. We can move some results back in, if the camera-ready version allows one additional page. Regarding the suggestion for additional ablation studies; unfortunately we cannot remove the discriminator as the discriminator loss is the one and only learning signal we backpropagate to the generator. Similarly, an ablation study of the role of the causal generator (vs a standard generator) would not be possible, as the causal generator is what allows us to debias. Note: though it is not exactly an ablation study, Appendices E and F include basic sensitivity analyses for respectively the choice of surrogate mechanism and the quality of the causal knowledge, thereby we hope these provide insight into how these two components of the model affect overall performance. We hope you might also be happy to learn that we are including additional experiments for higher-dimensional data, to highlight the method’s scalability.
>
> **[v. Surrogate variables]** In Section 6 we use mechanism I of Appendix E for the surrogate variables: we sample surrogate variables from the corresponding feature’s empirical marginal distribution (i.e., the training data).
>
> [1] Judea Pearl. Causality. Cambridge university press, 2009
>
> [2] Jonas Peters, Dominik Janzing, and Bernhard Schölkopf. Elements of causal inference: foundations and learning algorithms. p. 19. The MIT Press, 2017
>
> [3] Faisal Kamiran, Indre Žliobaite, and Toon Calders. Quantifying explainable discrimination and removing illegal discrimination in automated decision making. Knowledge and Information Systems, 2012
>
> [4] A. Borji, Pros and cons of gan evaluation measures, Computer Vision and Image Understanding (2019) 41–65.
>
> [5] M. Arjovsky, S. Chintala, and L. Bottou. Wasserstein gan. arXiv preprint arXiv:1701.07875, 2017.

---

> ### Author Response · Authors · 2021-08-25
> **Dear reviewer tL8r**
>
> Thanks again for reviewing our paper.
>
> Given the increased score of reviewer FVou, we were curious if there should remain any other concerns on your part. We believe that we have addressed the concerns mentioned in your review, and are happy to clarify further should you remain unsatisfied.
>
> Best, Authors of #4691

---

> ### Author Response · Authors · 2021-08-30
> **Dear reviewer tL8r**
>
> Dear reviewer tL8r,
>
> Given the short time left in this response period, we’d like to enquire whether our response addressed you concerns. If you found our response convincing, we kindly ask you to increase your score.
>
> Authors of #4691

---

### Official Review · Reviewer_xuZf · 2021-07-16

**Rating:** 6
**Confidence:** 2

**Summary:**

This work tackles the problem of fairness from a pre-processing perspective by handling the data first using some form of generative transformation. In particular, they use a GAN as this generation and embed a causal structure within the layers. There are then theoretical guarantees presented on the convergence and guarantees of fairness when using the processed data on other tasks.

**Limitations And Societal Impact:**

Yes they have.

**Main Review:**

The goal is the submission is to make work towards handling bias in machine learning systems but in particular take a pre-processing approach to this. Overall, such an approach has modularity since any high-performing or even biased algorithm can then use this method to produce fair ML systems, attesting to the significance of this approach.

The major strength of this submission is precisely this method due to its modularity, which as illustrated seems to be a practical method for practitioners. The weakness I see in this work are the theoretical guarantees, which seem to be quite implicit and lacking - for example, can the authors provide some kind of convergence rates as opposed to just guarantees? On first reading of 'convergence guarantees of the generator', I expected some form of distributional convergence or functional convergence (since they mention the generator will converge) so it would be better if the authors can be more candid about this contribution in the abstract. However, I do not think this is such a major concern.

I am not an expert in this area however the proposal of a GAN-based method that embeds causal structure seems to be quite original and the relevant work seems to be discussed. The paper is well-written with the relevant notation being self-contained and motivation spelled out clearly. I briefly checked the proofs and think the quality of correctness makes sense.

Overall, I rate this paper above the threshold due to the modularity of the approach however since it is not my expertise I cannot comment entirely on the novelty. I think this work is highly relevant to the NeurIPS community and certain can benefit practitioners. One question I had for the authors to better understand this work is: How would this method perform if you trained a GAN on the pre-processed data to create larger datasets?

**Time Spent Reviewing:**

8

---

> ### Author Response · Authors · 2021-08-10
> **Response to Reviewer xuZf**
>
> _Thank you for your time and thoughtful comments. We will try to address the two points that you raised, namely (i) on the convergence of the generator, and (ii) “How would this method perform if you trained a GAN on the pre-processed data to create larger datasets?”._
>
> **[i. Generator convergence]** Deriving convergence rates would be very interesting indeed, but for GANs this is notoriously hard. However, our guarantees follow the standard approach taken in much of the GAN-literature, e.g. the original GAN paper of Goodfellow et al. (2014)
>
> **[ii. Large datasets]** In fact one is able to choose how much data one generates with the DECAF GAN. We see this as an advantage over non-generative methods that only remove bias from the training data (e.g. [1]), as in that case the debiased dataset is necessarily equal in size to the original one (and one would indeed have to train another generative model on top of the new dataset for creating larger datasets). Thank you for this comment, we will mention this advantage in the main paper.
>
> If this does not answer your questions, or if you have any other questions that could help you in your assessment, please let us know.
>
> [1] Lu Zhang, Yongkai Wu, and Xintao Wu. A causal framework for discovering and removing direct and indirect discrimination. In Proceedings of the Twenty-Sixth International Joint Conference on Artificial Intelligence, IJCAI-17, pages 3929–3935, 2017.

---

> ### Author Response · Authors · 2021-08-25
> **Dear reviewer xuZf**
>
> Thanks again for reviewing our paper.
>
> Given the increased score of reviewer FVou, we were curious if there should remain any other concerns on your part. We believe that we have addressed the concerns mentioned in your review, and are happy to clarify further should you remain unsatisfied.
>
> Best,
>
> Authors of #4691

---

> ### Author Response · Authors · 2021-08-30
> **Dear reviewer xuZf**
>
> Dear reviewer xuZf,
>
> Given the short time left in this response period, we’d like to enquire whether our response addressed you concerns. If you found our response convincing, we kindly ask you to increase your score.
>
> Authors of #4691

---

### Official Review · Reviewer_SA2b · 2021-07-17

**Rating:** 6
**Confidence:** 3

**Summary:**

This work proposes a novel task of generating fair synthetic data by a GAN-based model. The generation process follows the order on a causal DAG and fairness is achieved by debiasing during the inference phase. Experiment results are carried out on Adult and Credit Approval datasets to show the effectiveness of generating fair synthetic data.


**Limitations And Societal Impact:**

I don’t think this work can lead to any negative social impact.


**Main Review:**

Originality:
- Generating fair synthetic data for machine learning is an interesting task. This paper designed a GAN-based framework which can achieve fair data generation by simply intervening the inference phase. The idea is novel and interesting.

Quality:
- Some discussion on efficiency: Although the idea of generating synthetic data following a topological order is an interesting idea, it also raises a concern about the efficiency of the algorithm. The algorithm needs to run the generator d times, where d is the number of columns in the table. If there are only a few protected columns, it’s possible to simplify the DAG by merging irrelevant columns. However, if the number of protected columns are also large, then there’s no easy fix to the efficiency.
- Comparing with more baseline methods: The method is only compared with GAN-based baselines. Since statistical methods are also capable of generating synthetic data, such baselines should also be included, for example PrivBayes [1]. Also, there are GANs (TableGAN [2], CTGAN [3]) and VAEs (TVAE [3]) designed specifically for synthetic table generation.
- The experiment is carried out on 2 small dataset, each with less than 20 columns. It’s unclear how the model would perform on a dataset with more columns.

Clarity
- This paper clearly defines different fairness in synthetic data. The high-level idea is also understandable.
- However, I would ask the authors to precisely describe the GAN model using either pseudo code or equations in the main paper, rather than burying all details deeply in the supplementary material and even in the source code. For example, the sharing weights in D.1 is important and quite hard to understand. It’s also unclear how the paper handles categorical (discrete) columns in tabular data.
- I also recommend the authors also provide stats for fair credit dataset.

Significance:
- This paper clearly lays out different definitions of fair synthetic data, and the proposed method is novel and interesting. These ideas are inspiring for future research.

[1] PrivBayes: Private Data Release via Bayesian Networks
[2] Data Synthesis based on Generative Adversarial Networks
[3] Modeling Tabular data using Conditional GAN


**Time Spent Reviewing:**

3

---

> ### Author Response · Authors · 2021-08-10
> **Response to reviewer SA2B**
>
> _Thank you for your comments and suggestions._
>
> **[i. Efficiency]** In the current form, indeed the run time scales linearly with the number of variables. In practice - for the larger Communities and Crime dataset (mentioned below) -  this comes down to an average training time of about 35s per epoch when run on a machine with hexacore Intel i7-6850K CPU. In our opinion, the linearly decreasing efficiency is thus not a problem in practice. Furthermore, practical improvements can be made to speed this up further: when the graph is sparse one can parallelize calculations and---like you have pointed out---one can cluster (some) variables and model these together.
> We agree that efficiency is an important consideration for many readers and we will include it in the discussion.
>
>
> **[ii. Baseline methods]** We will add additional baselines in the revised paper.  In the short-term (as a preview), we include an additional benchmark for VAE in the table below for Table 2 in the Census experiment.
>
> ╔════════╤══════════╤══════════╤══════════╤══════════╤══════════╗
>
> ║ Method     $\ \ \ \ \$       │ $\ \ \ \ \$     Precision  $\ \ \ \$   │ $\ \ \ \ \ \ \$   Recall   $\ \ \ \ \ \$          |  $\ \ \ \ \ \$   AUROC  $\ \ \ \ \$ │$\ \ \ \ \ \ \ \ \$ FTU   $\ \ \ \ \ \ \ \$      |  $\ \ \ \ \ \ \ \$ DP  $\ \ \ \ \ \ \ \  \$       ║
>
> ╠════════╪══════════╪══════════╪══════════╤══════════╪══════════╣
>
> ║ $\ \ \ \ \$ VAE   $\ \ \ \ \ \$   │ $\ \ \  $ 0.491 (0.03) $\ \ $│ $\ \  $ 0.454 (0.08) $\ \  $│ $\ \ \  $ 0.748 (0.01) $\ \ \  $ | 0.093 (0.01) | $\ \ \  $0.20 (0.01)  $\ \ $║
>
> ╚════════╧══════════╧══════════╧══════════╧══════════╧══════════╝
>
> **[iii. High-dimensional data]** Thank you for pointing out this concern.  We have put together an experiment on the Communities and Crime dataset ([38] in the original paper), which include features such as population demographics, law enforcement per capita, median income, etc.  The dataset contains 128 attributes and 1994 samples.  Using the same experimental setup from Section 6.1 for **Debiasing Census Data**, we repeat our experiments for this larger dataset. Here we predict the median income, and assume that one of the ethnic demographics is a protected variable.  After data cleaning the resulting DAG had 101 nodes with over 1200 edges.  We show some preliminary results in the table below, and will be sure to provide the full table and additional details in the revised manuscript.  We hope that this convincingly demonstrates that DECAF can scale to even larger datasets.
>
> ╔═══════════╤═══════╤════════╤════════╤════════╗
>
> ║ $\ \ \ \ \ \$Method  $\ \ \ \ \ \ $      │ $\ \ $Precision$\ $   │ $\ \ \ $Recall$\ \ \ $      │ $\ \ \ \ \ $FTU $\ \ \ \ \ \ $        |  $\ \ \ \ \ \ $DP $\ \ \ \ \ \ $         ║
>
>
> ╠═══════════╪═══════╪════════╪════════╪════════╣
>
> ║ Original data D   │ 0.932 (0.04)│ 0.897 (0.03)│ 0.14 (0.10) | 0.122 (0.03) ║
>
> ║ GAN      $\ \ \ \ \ $ $\ \ \ \ $ $\ \ \ \ $  │ 0.632 (0.10)│ 0.421 (0.05)| 0.09 (0.17) | 0.087 (0.01) ║
>
> ║ WGAN-GP      $\ \ \ \ \ \ \$     │ 0.687 (0.03)│ 0.346 (0.03)| 0.10 (0.18) | 0.122 (0.09) ║
>
> ║ FairGAN  $\ \ \ \  \ \ \ \ $         │ 0.539 (0.10)| 0.616 (0.04)| 0.18 (0.05) | 0.229 (0.15) ║
>
> ║ DECAF-FTU  $\ \ \ \ $       │ 0.730 (0.07)| 0.513 (0.04)| 0.02 (0.01) | 0.105 (0.05) ║
>
> ║ DECAF-DP    $\ \ \ \ \ $      │ 0.770 (0.02)| 0.496 (0.05)| 0.02 (0.01) | 0.043 (0.01) ║
>
> ╚══════════╧════════╧═══════╧═══════╧═════════╝
>
>
> **[iv. GAN inner workings]** We apologize for the confusion and would like to address the points you raised concerning clarity and the the inner workings of the GAN:
>
> Regarding the sharing of layers of the generator: each generator $G_i$ is an MLP, constructed by composition of two unique single layers (first and last) and shared central layers. Specifically, for some hidden dimension size $d_h$, we write $G_i = h_i \circ g \circ f_i$ with $f_i:\mathbb{R}^{|Pa(X_i)|+1} \rightarrow \mathbb{R}^{d_h}$ and $h_i:\mathbb{R}^{d_h}\rightarrow 1$ the single layer MLPs, for all $i$, and $g: \mathbb{R}^{d_h}\rightarrow \mathbb{R}^{d_h}$ shared.
> Regarding discrete variables, in both datasets the only non-binary discrete variable is the protected attribute, which we binarised ourselves (discriminated vs non-discriminated). All variables are generated in the same way, but binary variables are rounded off after generation.
> We will include these details in the main paper as you suggested.
>
> **[v. Stats of Credit Approval dataset]** We will include the stats for both datasets (as well as the new *Communities and Crime* dataset)  in Appendix D.3. Namely:
>
> ╔════════════════════╤════════╤════════╤══════════╗
>
> ║   $\ \ \ \  \ \ \ \ \ \ \ \ \ \ \ \ \ \ \ \ \ \ \ \ \ \ \ \ \ \ \ \ \ \ \ \ \ \ \ \ \ \ \ \ \ $ │ $\ \ \ $  Credit $\ \ \ $ │  $\ \ \ $ Census $\ \ \ $ | Communities ║
>
> ╠════════════════════╪════════╪════════╪══════════╣
>
> ║ Number of features     $\ \ \ \ \ \ $  $\ \ \ \ \ \ \ $          │  $\ \ \ \ \ \ $  15 $\ \ \ \ \ \ $    │ $\ \ \ \ \ \ $   10 $\ \ \ \ \ \ $   | $\ \ \ \ \ \ \ $   128  $\ \ \ \ \ \ \ $    ║
>
> ║ - Continuous           $\ \ \ \ \ \ $$\ \ \ \ \ \ $  $\ \ \ \ \ \ $ $\ \ \ \ $    │ $\ \ \ \ \ \ $   3 $\ \ \ \ \ \ \ \ $      │ $\ \ \ \ \ \ \ \ $   4$\ \ \ \ \ \ \ \ $     |  $\ \ \ \ \ \ $  120 $\ \ \ \ \ \ \ \ $     ║
>
> ║ - Discrete            $\ \ \ \ \ \ $$\ \ \ \ \ \ $  $\ \ \ \ \ \ $ $\ \ \ \ \ \ \ \ \ $                 │  $\ \ \ \ \ \ $    12  $\ \ \ \ \ \ $     │  $\ \ \ \ \ \ \ \ $  6 $\ \ \ \ \ \ \ $    |     $\ \ \ \ \ \ \ \ \ $8 $\ \ \ \ \ \ \ \ \ $      ║
>
> ║ Target type         $\ \ \ \ \ \ $$\ \ \ \ \ \ $  $\ \ \ \ \ \ $  $\ \ \ \ \ \ $                  │ $\ \ \ $    Binary $\ \ \ $   │ $\ \ \ $ Binary $\ \ \  \ $ | $\ \ \ \ $   Binary $\ \ \ \ $  ║
>
> ║ Number of samples     $\ \ \ \ \ \ $  $\ \ \ \ \ \ $           │  $\ \ \ \ \ \ $    679 $\ \ \ \ \ $     │ $\ \ \ $ 32,561 $\ \ \ $ |    $\ \ \ \ \ \ $1994 $\ \ \ \ \ \ \ $    ║
>
> ║ Number of discovered edges         │ $\ \ \ \ \ \ $    40 $\ \ \ \ \ \ $       │ $\ \ \ \ \ \ $     22 $\ \ \ \ \ \ \ $     |   $\ \ \ \ \ \ $ 1288  $\ \ \ \ \ \ $   ║
>
> ╚════════════════════╧════════╧════════╧══════════╝

---

> ### Author Response · Authors · 2021-08-25
> **Dear reviewer SA2b**
>
> Thanks again for reviewing our paper.
>
> Given the increased score of reviewer FVou, we were curious if there should remain any other concerns on your part. We believe that we have addressed the concerns mentioned in your review, and are happy to clarify further should you remain unsatisfied.
>
> Best,
> Authors of #4691

---

> ### Author Response · Authors · 2021-08-30
> **Dear reviewer SA2b**
>
> Dear reviewer SA2b,
>
> Given the short time left in this response period, we’d like to enquire whether our response addressed you concerns. If you found our response convincing, we kindly ask you to increase your score.
>
> Authors of #4691

---

> ### Comment · Reviewer_SA2b · 2021-09-01
> **Changed my score 5 -> 6**
>
> Thanks for providing additional experimental results regarding my questions.

---

> > ### Author Response · Authors · 2021-09-02
> > **Thank you!**
> >
> > Thank you for increasing your score and for getting back to us!  We will be sure to include all of the changes in the revised manuscript!

---

### Official Review · Reviewer_FVou · 2021-07-19

**Rating:** 6
**Confidence:** 4

**Summary:**

This paper proposes a GAN based causal aware generative model to generate fair data for downstream models. They provide theoretical motivation and practical solutions with different causal criteria.

**Limitations And Societal Impact:**

This paper is on the topic of responsible AI and if successful. It brings a more positive impact to machine learning applications. There is no explicit discussion regarding the potential negative societal impact. But I do encourage the authors to discuss more if people rely on their method (which may contain error from causal graph or model training), how to communicate to people about the results, and fairness guarantees. Also, consider how to communicate the assumptions to users to make sure that the methods are used correctly.

**Main Review:**

Pros:
* This paper considers using a causal aware generative model to generate fair datasets, which I believe is the right approach to handle this problem.
* The paper is presented clearly
* The paper provides a theory to motivate the proposed solution
* The paper has a meaningful discussion regarding how the method can be used in practice (tool for discovery and markove compatible)

Cons:
* Concerns regarding stability of the method (2 below)
* Concerns about biased data leads to wrong DAG, which cannot generate fair data (point 5 below)

5 is my main concern below comparing to others, and I am looking forward to your reply and happy to change the score if 5 can be addressed.

Details:

1. You seem need to assume to know the underlying downstream task. If the task selects a different variable as a target, the fairness guarantee won't be there anymore, right? Do you have any thoughts on generating fair datasets generation?
2. I have concerns about the scalability of the method. If it is tabular data with many variables and with many edges, how would you implement it? ALso, do you need to write a new piece of code given any new causal graph or do you automate it in some way? If so, how?
3. Causal graph is known is a strong assumption. But it is fine as you discussed discovery methods. In this case, do you have results for your method with respect to error from causal discovery?
4. line 208. You mentioned Markov compatible and in experiments, you used Tetrad. I assume that the output from the Tetrad with score-based method will be a CPDAG. How did you use CPDAG to design the generator? Did you just take one sample from the Markov equivalent class. How robust is your method to different DAG in the Markov equivalent class? Does these make a difference in experiments?
5. I believe that causal sufficiency is an assumption here.  However, bias in the dataset itself may be due to selection bias. And existing causal discovery methods have problems to hand selection bias. Thus, the root of the bias cannot to resolved. (the root cause of unfair also it the root cause of the failure of causal discovery methods. ) For example, in 20s the survey about IQ, gender etc where there is fewer females in university where the survey is taken and leads to wrong conclusions. Such selection bias makes causal discovery fail and also makes other ML methods unfair. [Check causal discovery for MNAR papers]
6. From the generative modeling point of view, the model seems just the GAN version of the CAMA model Zhang C, Zhang K, Li Y. A Causal View on Robustness of Neural Networks. Advances in Neural Information Processing Systems. 2020;33.

**Time Spent Reviewing:**

3

---

> ### Author Response · Authors · 2021-08-10
> **Response to Reviewer FVou**
>
> _Thank you for your comments and suggestions. We address your main concern first (item 5), then proceed through the rest of the comments in order they were presented._
>
> **[v. Causal sufficiency]** Thank you for this important comment and highlighting this as your main concern. We indeed assume causal sufficiency, as is implied by Eq. (1) with $Z_i$ independent (but we will state this explicitly in the paper). Causal sufficiency is common in causal (fairness) literature (e.g. [1,2]), but we acknowledge that this is an important limitation of the method---see also the Discussion (line 346). However, any other method that attempts to address fairness notions based on causal relations (e.g. [1-4]), is also limited by the causal knowledge that underlies the fairness specification, which in turn is dependent on the success of the causal discovery method. As such, we see this not as a limitation of just the proposed method, but as a limitation inherent to all causal fairness and discovery literature.
>
> Note that if we assume full causal graphical knowledge including hidden variables, the effect of the hidden variable (e.g. selection bias) can sometimes be accounted for, e.g. using a latent-variable model [5]. This is beyond the scope of this work.
>
> On a positive sidenote, many definitions of fairness including the three definitions of FTU, DP and CF in the main paper, are correlation based and do not explicitly need (full) causal knowledge. Though our method uses a causal graph to generate data, as also discussed in point (iv) below, we prove (Theorem 2) that in fact this graph needs to only be Markov compatible with the (observational) distribution for guaranteeing the generator’s convergence. More importantly, an incorrect graph can negatively affect the quality of samples, but not the FTU, DP or CF fairness guarantees as long as the graph reflects the relevant conditional dependencies, see Corollaries 1-3. This is reflected in the experimental results; it is not unlikely the two real-world datasets in fact contain hidden confounders, yet debiasing is successful. We will include another experiment on the Credit dataset with hidden confounders: we will create a hidden confounder by removing the variable “Default” (see Figure 3 in Appendix D), re-run the causal discovery method and otherwise replicate Experiment 6.2.
>
> If we understand the comment about causal discovery in the MNAR missingness setting correctly, you suggest framing the hidden confounder setting in terms of the missing data framework [6], specifically as an MNAR setting. We would like to stress that we assume no missingness in training nor test data. Even if one would allow for hidden variables, in our opinion the missingness framework (and MNAR) would not be an appropriate framework as we would assume a variable to be either always observed or always hidden. In contrast to this, causal discovery methods with MNAR hidden confounders observe the hidden variables for some samples.
>
> As you mentioned that this is your main concern, please do let us know if this response does not address all your questions or if you have any other questions.
>
>
>
> **[i. Different target]** It is correct that we need to know the target(s) of the downstream task, because all definitions of fairness are defined with respect to a protected attribute and target variable. That being said, multiple datasets can be created and released with different (sets of) targets and/or protected attributes. This would not require re-training and can be done during inference time, which we see as a big advantage of our method (see Remark, line 246). We will explicitly state the importance of knowing the target variable.
>
>
> **[ii. Scalability]** Regarding generalization and extensibility of DECAF to arbitrary or new causal DAGs, the algorithm is implemented such that this is not an issue. In the DECAF source code we provided, you will notice that DECAF requires two inputs.  The first is the DAG provided/represented as a binary adjacency matrix. The second required input is the edges we wish to remove, which we represent as items in a hash map.  We provide this at inference time to DECAF.  This also allows us to generate multiple datasets with different fairness restrictions - by tuning the edges we include in the input *edge restriction* dictionary.  Please note that this *does not* require us to write new code, and only set inputs into DECAF.  Regarding the scalability of DECAF to many variables, fortunately this is in practice not a limitation. Here we include a high-dimensional experiment:
>
> With a total run time of about 35s per training epoch on the *Communities and Crime* dataset (which has 100+ features and over 1200 edges). The larger limiting issue would be identifying large enough DAGs.  In certain scenarios some DAGs may be more time consuming to train than others. Conversely, there may be some DAGs that are extremely parallelizable, and could benefit from parallelized or distributed training. Furthermore, in some cases one can cluster together multiple variables and model these together as a single node. We reserve this analysis for future work, and will clarify this as a potential limitation in the Discussion.
>
>
> **[iii. Error DAG]** Indeed, knowing the complete causal DAG is a strong assumption, and is likely violated when the number of variables increases. In Appendix F we have included experiments for incorrect DAG specifications on the Credit Approval dataset, to test for its effect on data quality and fairness.
>
> _(Quality)_ We observe that the quality (i.e. precision, recall, AUC) deteriorates when certain edges are absent from the assumed DAG. The same is observed when too many edges are included; in this case the DAG is no longer faithful w.r.t. the underlying distribution, which means that too many edges might be removed for debiasing (specifically, Theorem 1 does not hold), leading to quality degradation.
>
> _(Fairness)_ Overall the results in Appendix F suggest that the ability to debias is not significantly affected by an incorrect DAG. Note first that adding edges can only induce additional dependencies, thus from Proposition 1---and by extent Corollaries 1-3---adding edges does not affect our fairness guarantees. Removing---and to a lesser extent, reversing---edges means Proposition 1 no longer holds, as we might wrongly consider $A\perp_{\mathcal{G}} B | R$. However, the experimental results indicate fairness remains largely unaffected.
>
> **[iv. Markov equivalence]**  You are correct that we did take one sample from the Markov equivalence class (MEC), which indeed we should have mentioned in the paper. We observed in our experiments that the DAGs in the MEC performed similarly.  This makes sense because all graphs in the MEC are Markov compatible with the true data distribution, and from Theorem 2 it follows that any graph in the MEC is able to give a valid generation order for the generator. See also the brief discussion on Markov equivalence classes at the end of Appendix B (line 56). Please let us know if this addresses your concerns, otherwise we are happy to provide additional results for different graphs in the MEC.
>
> **[vi. On CAMA]** In our paper, we propose a solution to providing fair (synthetic) data for any downstream learner. For this we present a GAN-framework that learns, and infers, using the topological structure of a causal diagram. While there are indeed parallels to Zhang et al., our focus is entirely different. With CAMA, Zhang et al. provide a causal perspective on robustness for neural networks in general. Here, robustness is defined in terms of adversarial examples, i.e. robustness against out-of-sample observations. Accordingly, the experimental results provided by Zhang et al. mainly focus on classification, not data generation. We agree, however, that Zhang et al. is relevant to our work and will cite and discuss it appropriately in our main text.
>
>
> **[Potential negative societal impact]** Thank you for this recommendation. We will highlight, as discussed above, 1) how the quality of data is reliant on the DAG discovery and causal sufficiency assumption, and 2) how fairness is only guaranteed with respect to the chosen target. In general, we regard the potential for negative societal impact low, considering that both data quality and fairness (given a correct definition) can always be experimentally validated on a hold-out test set prior to public release.
>
>
>
> [1] Lu Zhang, Yongkai Wu, and Xintao Wu. A causal framework for discovering and removing direct and indirect discrimination. In Proceedings of the Twenty-Sixth International Joint Conference on Artificial Intelligence, IJCAI-17, pages 3929–3935, 2017.
>
> [2]  Niki  Kilbertus,  Mateo  Rojas  Carulla,  Giambattista  Parascandolo,  Moritz  Hardt,  Dominik Janzing,  and Bernhard Schölkopf.   Avoiding discrimination through causal reasoning.   InI. Guyon, U. V. Luxburg, S. Bengio, H. Wallach, R. Fergus, S. Vishwanathan, and R. Garnett,  editors, Advances  in  Neural  Information  Processing  Systems,  volume  30.  Curran Associates,  Inc.,  2017.
>
> [3] Razieh Nabi and Ilya Shpitser. Fair inference on outcomes. In Proceedings of the AAAI Conference on Artificial Intelligence, volume 32, 2018.
>
> [4] Russell, C., Kusner, M., Loftus, J., and Silva, R. When worlds collide: integrating different counterfactual assumptions in fairness. Advances in Neural Information Processing Systems, 31, 2017.
>
> [5] Christos Louizos, Uri Shalit, Joris M. Mooij, David Sontag, Richard Zemel, Max Welling. Causal Effect Inference with Deep Latent-Variable Models. Advances in Neural Information Processing Systems, 31, 2017.
>
> [6] Roderick J. A. Little, Donald B. Rubin. Statistical Analysis with Missing Data (2nd ed.), Wiley, 2002.

---

> > ### Comment · Reviewer_FVou · 2021-08-25
> > **5->6**
> >
> > I have read the author response and changed my score to 6 assuming that the author will revise the paper and discuss more clearly about the limitations as promised.

---

> > > ### Author Response · Authors · 2021-08-25
> > > **Thank you!**
> > >
> > > Dear reviewer FVou,
> > >
> > > Thank you for increasing your score.
> > >
> > > We agree to include the above discussion regarding the limitations of DECAF into our main text.

---

### Author Response · Authors · 2021-08-10
**General Response**

We express our sincerest gratitude for taking the time to read and evaluate our manuscript.  We hope our responses satisfy the concerns raised, and should there still remain points that are unclear, please do not hesitate to ask for additional clarification as we are happy to respond to additional inquiries.

---

### Author Response · Authors · 2021-08-21
**Additional Comments**

We want to reiterate our sincerest gratitude to all the reviewers of our paper. If you have any remaining concerns, please let us know - we would be happy to do our utmost to address them!

---

### Author Response · Authors · 2021-09-02
**General response**

We would like to thank all reviewers once again. We believe our responses have addressed all comments and concerns, and we will make sure your feedback will be reflected in the final version of the paper. We regard fair synthetic data as a promising new avenue into fair machine learning---and an idea that is currently underexplored in the synthetic data community (e.g. not included in *Tutorial on Synthetic Data*, ICML 2021)---therefore we hope that the paper will be included in NeurIPS 2021.

Best wishes,

Authors of #4691

---

### Decision · Program_Chairs · 2021-09-27

**Decision:**

Accept (Poster)

**Comment:**

While the paper has initially some disagreement in score, it now arrives at a unanimous consensus by properly addressing all of the main concerns, in particular, regarding the limitation of the underlying assumption made in the paper, as well as providing more experimental results w.r.t. suggested baselines. I believe the paper is worth being published as a poster.